# Student-Centered Distillation Narrows the Agentic Gap Between Small and Large LLMs

Yuanjie Lyu [1]   Chengyu Wang [2]   Jun Huang [2]   Tong Xu [1]

## Abstract

Large Language Model agents achieve strong performance on multi-step reasoning and tool-use tasks, but their impressive capabilities typically rely on extremely large backbones. Existing distillation approaches train smaller students to imitate full teacher trajectories, yet reasoning and knowledge gaps between the teacher and student can cause compounding errors. We propose *SCoRe*, a student-centered framework in which the student generates training trajectories and the teacher corrects only the earliest error, producing training data matched to the student's abilities and exposing specific weaknesses. The student is first fine-tuned on corrected trajectories. Subsequently, short-horizon reinforcement learning starts from the verified prefix preceding the earliest error, with target rewards assigned at that step. This design enables the student to solve problems through unconstrained RL exploration rather than teacher imitation, while the short-horizon setup improves training stability. On 12 challenging benchmarks, a 7B-parameter student distilled with *SCoRe* closes the agentic performance gap with a 72B-parameter teacher[1].

## 1. Introduction

Recent advances in Large Language Models (LLMs) have led to the rise of "agents" (Xi et al., 2025). Unlike traditional single-pass generation, LLM agents solve complex problems through an iterative *reasoning–action–observation* loop, using frameworks such as ReAct (Yao et al., 2023) and CodeAct (Wang et al., 2024). Specifically, LLM agents decompose tasks into sub-goals (reasoning), execute them via external tools such as code interpreters (action) (Schick et al., 2023; Gao et al., 2023), and then refine their plans based on feedback from tool execution (observation). By combining LLM planning with the precision of external tools, agents mitigate flaws of LLMs such as hallucinations and outdated knowledge, achieving strong performance on real-world interactive tasks (Liu et al., 2024).

High-performing agents typically use large backbones such as GPT-series (Achiam et al., 2023), resulting in high latency and cost, as complex tasks may require dozens of model calls. To reduce deployment cost, Agent Distillation (Kang et al., 2025) decomposes the teacher model's behavior into structured [`Thought`, `Action`, `Observation`] trajectories and trains a smaller student model to imitate them. This enables the student to acquire part of the teacher's reasoning ability while reducing cost. However, this imitation faces two main challenges: (1) *Reasoning Ability Gap:* smaller models often fail to reproduce the teacher's logical decomposition (Magister et al., 2023); (2) *Knowledge Capability Gap:* the student may be unable to execute complex actions despite following the teacher's plan. Both gaps stem from the *emergent abilities* (Wei et al., 2022) of LLMs, gained through extensive training and not fully transferable to smaller models. Moreover, in behavior cloning, a single failure at any step can push the student into out-of-distribution states, where errors compound and grow as $O(H^2)$ with horizon length $H$ (Ross et al., 2011).

To address the limitations of the traditional "teacher-acts, student-clones" distillation paradigm, we introduce *SCoRe* (**S**tudent-**C**entered **O**ne-step **Re**inforcement). *In SCoRe, the student first produces a complete trajectory, which the teacher reviews globally, intervening only to correct the earliest error*. The student then restarts the task, resuming from the original correct prefix and the teacher's correction. As shown in Figure 1, this student-centered approach to data generation offers two key advantages: *Capability Matching*, where trajectory complexity aligns with the student's abilities, keeping the data learnable and effective; and *Deficiency Localization*, where the structure of a "verified prefix" followed by a "key step" explicitly pinpoints the student's weaknesses for targeted optimization.

---

[1]University of Science and Technology of China, Hefei, China [2]Independent Researcher (in personal capacity). Correspondence to: Chengyu Wang <chywang2013@gmail.com>, Tong Xu <tongxu@ustc.edu.cn>.

*Proceedings of the 43$^{rd}$ International Conference on Machine Learning*, Seoul, South Korea. PMLR 306, 2026. Copyright 2026 by the author(s).

[1]Code: github.com/haruhi-sudo/SCoRe

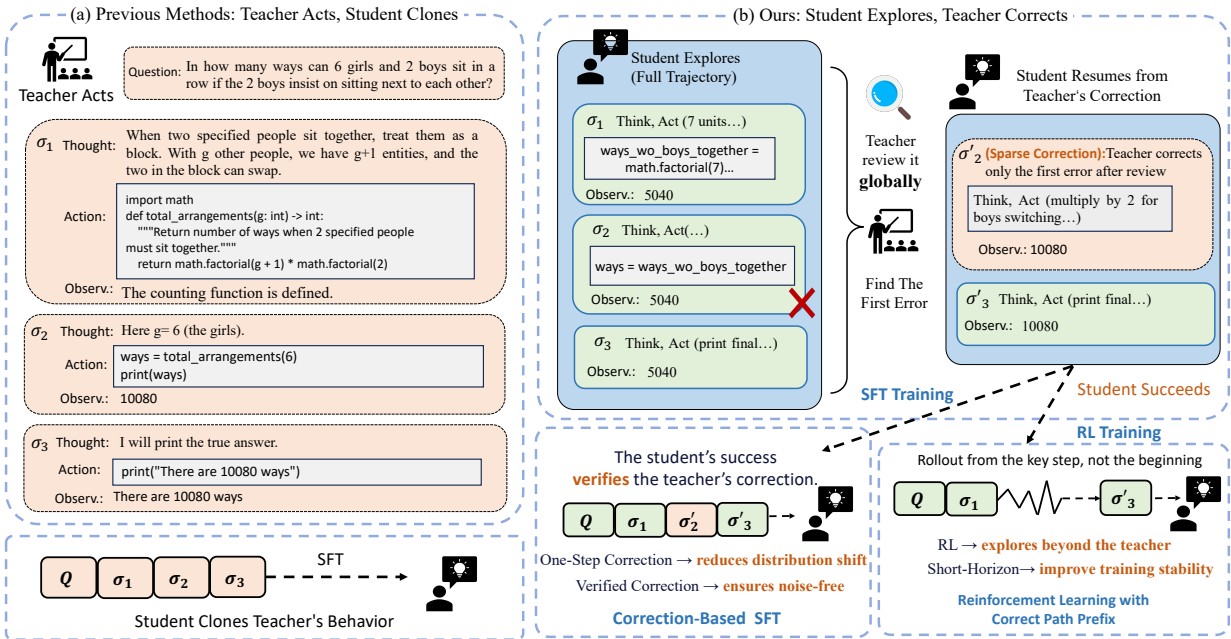

*Figure 1.* Comparison between imitation-based distillation and our *SCoRe* framework. (a) Prior methods clone entire teacher trajectories. (b) Our approach lets the student explore independently; *upon completion, the teacher reviews the full trajectory* and minimally intervenes by correcting only the earliest error. Correction-based SFT mitigates the compounding errors of pure imitation. RL rollouts then start from this verified prefix, improving stability and efficiency.

Before exploration, the student must first acquire basic reasoning–action skills. We initialize it with a cold-start *Behavior Cloning* (BC) phase, performing supervised fine-tuning (SFT) on a small set of high-quality teacher trajectories. Then, the core phase, *Mentored Problem-Solving* (MPS), allows the student to tackle new tasks independently. After the student completes a trajectory, a teacher inspects it, identifies and corrects the earliest error, and the student regenerates the remainder starting from this corrected prefix. If further mistakes occur, the teacher repeats this process. Final task success implicitly verifies the correctness of the teacher's fix[2]. We retain these corrected trajectories with minimal teacher intervention and use them for the next round of SFT. By correcting the earliest error, this approach limits teacher–student distribution shift to a single step, breaks long error chains of BC, and yields $O(H)$ cumulative error growth rather than $O(H^2)$.

Although effective, these steps still rely on teacher corrections and keep the student in imitation mode. To promote genuine problem-solving and even enable the student to surpass the teacher, we introduce a reinforcement learning (RL) phase featuring shorter rollouts and key-step rewards. Our RL method introduces two main innovations. First, rather than rolling out from the start of the task, we begin from the verified prefix preceding the student's original error. This shortens the horizon and reduces gradient-estimation variance, making updates more stable. Second, to mitigate the issue of sparse rewards (Andrychowicz et al., 2017), we supplement the final task-success reward with additional bonuses at key steps: a larger one for reproducing the teacher's correction and a smaller format reward for avoiding the original error. Finally, during the MPS phase, we retain the data that remains unsolved despite multiple one-step corrections. A small part of this data is marked as challenging and used in RL training.[3]

Experiments on 12 challenging benchmarks show that *SCoRe* enables small models, such as Qwen2.5-7B-Instruct, to achieve tool-use performance comparable to that of a 72B teacher, outperforming both BC (Torabi et al., 2018) and GRPO (Shao et al., 2024) by a large margin. These results demonstrate that ability-matched correction, combined with short-horizon key-step RL, can close the performance gap between small and large models.

## 2. Related Work

**Distillation for LLM Agents.** Agent distillation is a practical way to reduce the cost of deploying large models such as GPT-4 (Achiam et al., 2023) or Gemini (Team et al.,

---

[2]Every teacher correction should pass this verification to reduce the risk of noisy corrections.

[3]RL optimizes through exploration rather than imitation; the training signal is not fully determined by the teacher, and the policy is therefore not bounded by the teacher.

2023) in a multi-turn agent setting (Yao et al., 2023; Wang et al., 2024). Formal approaches such as trajectory distillation (Kang et al., 2025) and Toolformer-style imitation (Schick et al., 2023; Gao et al., 2023) train smaller models to reproduce teacher-generated `[Thought, Action, Observation]` traces, transferring planning and tool-use skills. However, most rely on supervised behavior cloning from expert rollouts, making them vulnerable to teacher–student reasoning and knowledge gaps (Magister et al., 2023; Cai et al., 2025) and to compounding errors under distribution shift (Ross et al., 2011).

DAgger and HG-DAgger (Ross et al., 2011; Kelly et al., 2019) aim to mitigate compounding errors; however, these methods are designed for traditional imitation learning domains such as robotic control, and they are not directly applicable to LLM agents, which autonomously employ multiple tools. Furthermore, they remain strictly teacher-led and do not leverage a stable RL framework to progressively expand the student's capabilities.

**Agentic Reinforcement Learning.** RL has become a core paradigm for training LLM agents in dynamic, multi-turn environments (Shridhar et al., 2021; Mialon et al., 2024). Early pipelines relied on supervised or rule-based tool-use strategies (Schick et al., 2023), limiting adaptability to new domains. Later work integrated RL into agent policies, enabling joint optimization of reasoning chains and external actions. Classical methods such as deep Q-learning (Mnih et al., 2015) and self-play (Silver et al., 2017) have been adapted for natural-language decision-making, where intermediate thoughts and tool calls form part of the state (Li et al., 2025a). In tool-augmented settings, recent work has optimized multi-tool coordination under real-time constraints (Qian et al., 2025; Xu & Peng, 2025). Despite these advances, challenges remain: long-context rollouts cause instability (Schulman et al., 2017; Peng et al., 2019), and sparse or delayed rewards hinder credit assignment (Andrychowicz et al., 2017). These challenges motivate algorithms that combine trajectory-level planning with local, verifiable corrections for stable learning and fine-grained reasoning control.

Closely related work is further discussed in Appendix A.

## 3. Background

We formalize the LLM-agent framework and review two training methods: Behavior Cloning and Reinforcement Learning, highlighting their limitations in agent distillation.

**The ReAct Framework.** To enable effective interaction with an external environment $\mathcal{E}$ (e.g., a code interpreter or a search engine), we adopt the ReAct framework (Yao et al., 2023). In this framework, the agent's behavior is represented as a trajectory $\tau = (t_1, c_1, o_1, \ldots, t_H, c_H, o_H)$,

where $H$ is the total number of steps. At each step $i$, the policy $\pi$ first generates a *thought* $t_i$ based on the history $s_i$, and then, conditioned on this thought and history, produces a ReAct *action* $c_i$:

$$s_i = (t_1, c_1, o_1, \ldots, t_{i-1}, c_{i-1}, o_{i-1}), \quad (t_i, c_i) \sim \pi(\cdot \mid s_i).$$

Executing $c_i$ in $\mathcal{E}$ returns an *observation* $o_i$, and $(t_i, c_i, o_i)$ is then added to the history. The loop continues until a terminal action (e.g., producing the final answer) ends the trajectory.

**Agent Distillation via Behavior Cloning.** To reduce the inference cost of large LLM agents, agent distillation transfers capabilities from a powerful teacher $\pi_E$ to a smaller student $\hat{\pi}$. The most common method, *Behavior Cloning (BC)* (Torabi et al., 2018), trains $\hat{\pi}$ on teacher-generated trajectories $D_T$ to imitate both thoughts $(t_i)$ and ReAct actions $(c_i)$:

$$\mathcal{L}_{\text{BC}}(\theta) = -\mathbb{E}_{\tau \sim D_T} \left[ \sum_{i=1}^{|\tau|} \log \hat{\pi}(a_i \mid s_i; \theta) \right], \quad (1)$$

where $a_i = (t_i, c_i)$ and $s_i$ is the history context. While BC is effective, *reasoning* and *knowledge* gaps between $\pi_E$ and $\hat{\pi}$ yield a non-trivial per-step error rate $\varepsilon$. Under covariate shift, these errors accumulate over the horizon $H$, making the expected total cost grow as $O(H^2\varepsilon)$ (Ross et al., 2011).

**Agent Optimization via Reinforcement Learning.** Reinforcement Learning (RL) extends beyond imitation by enabling agents to explore actively. The problem is modeled as a Markov Decision Process (MDP), where the state $s_i$ is the history context, the action is $a_i$, the policy is $\hat{\pi}$, and the trajectory is evaluated by a reward $R(\tau)$. In practice, rewards are often sparse; for example, $R(\tau) = 1$ if the final answer is correct, and $R(\tau) = 0$ otherwise (Su et al., 2025; Mroueh, 2025). The training objective is $J(\theta) = \mathbb{E}_{\tau \sim \hat{\pi}_\theta}[R(\tau)]$, with the policy gradient update:

$$\nabla_\theta J(\theta) = \mathbb{E}_{\tau \sim \hat{\pi}_\theta} \left[ R(\tau) \nabla_\theta \log P(\tau \mid \theta) \right], \quad (2)$$

where $P(\tau \mid \theta)$ denotes the probability of trajectory $\tau$ under $\hat{\pi}_\theta$. Standard RL faces challenges for LLM agents due to *sparse rewards* (Andrychowicz et al., 2017; Badia et al., 2020) and *high variance* (Schulman et al., 2017; Shao et al., 2024), often leading to unstable and inefficient training.

## 4. Methodology

In this work, we propose *SCoRe*, a principled student-centered distillation framework that addresses key limitations of BC and RL (Figure 2). Instead of the static "teacher-acts, student-clones" scheme, *SCoRe* introduces *Mentored Problem-Solving* (MPS) to place the student at the center of learning. Training on MPS-generated data tailored to the

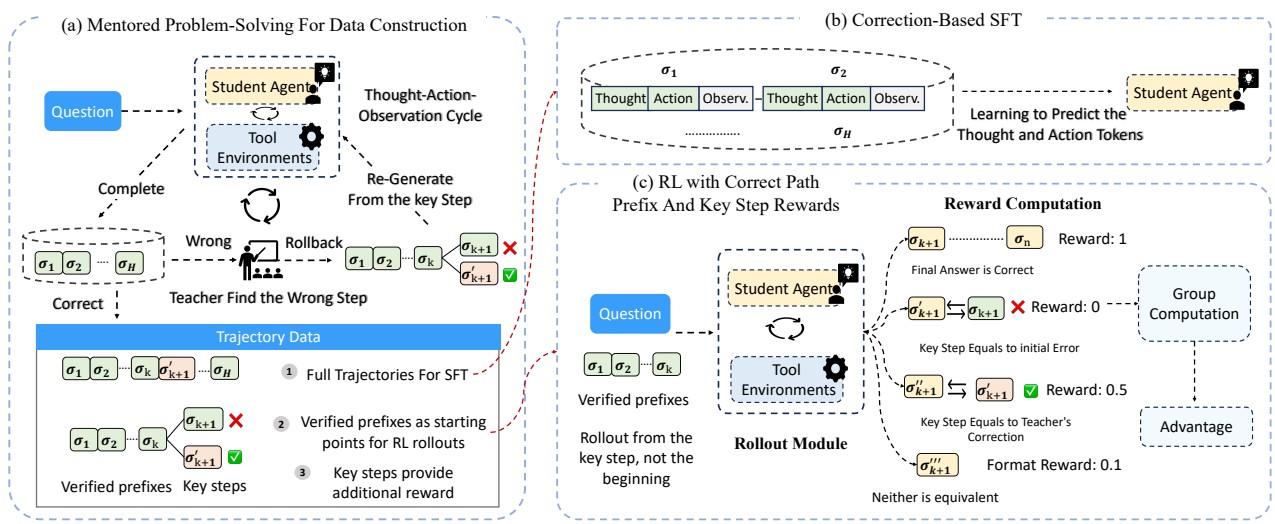

*Figure 2.* The *SCoRe* framework. (a) A student agent attempts a task, and the teacher provides a single-step correction at the first error, creating student-centered training data. (b) The student is initially trained to imitate full solution trajectories via supervised fine-tuning. (c) The student is further improved through reinforcement learning, using shortened rollouts starting from the prefix preceding the teacher's correction and targeted rewards at the corrected steps to guide exploration.

student's abilities reduces BC's compounding error growth, and the use of short-horizon, key-step rewards in RL improves stability and efficiency.

## 4.1. Initial Distillation with Code as Action

To enable a "student-explores, teacher-corrects" strategy for agent distillation, the student must first acquire basic *reasoning-acting* skills. However, some LLMs (e.g., Qwen2.5 (Yang et al., 2025), LLaMA3 (Grattafiori et al., 2024)) are not built for multi-step agentic tasks and typically perform single-pass generation. Therefore, we first construct structured datasets and distill $\hat{\pi}$ to perform multi-step reasoning and acting.

**Trajectory Representation.** Training trajectories follow the ReAct (Yao et al., 2023) format: sequences of triplets $\tau = (\sigma_i)_{i=1}^H$, $\sigma_i = (t_i, c_i, o_i)$, where $t_i$ is textual reasoning, $c_i$ is an action in ReAct, and $o_i$ is environment feedback. These structured trajectories capture problem decomposition and stepwise solutions, making them well-suited for distillation purposes. We build on *CodeAct* (Wang et al., 2024), where $c_i$ is executable code, offering: (i) deterministic operations for reproducibility; (ii) both the teacher and the student are familiar with code from pre-training, reducing capability gaps; and (iii) a Turing-complete, unified action space for tool use and complex logic.

**Initial Trajectory Generation.** To build the initial training data $D_T$, we use the teacher policy $\pi_E$. Following (Kang et al., 2025), $\pi_E$ is first prompted to produce a high-level plan `<first_thought>` as a strategic outline. A second prompt enforces the standard *Thought–Code–Observation*

cycle: at step $i$, $\pi_E$ generates $(t_i, c_i)$ conditioned on both the plan and prior steps. Executing $c_i$ in the environment yields an observation $o_i$, which is appended to the context for subsequent reasoning. This dual-prompt design combines global planning with adaptive step-level reasoning. To ensure quality, we apply rejection sampling and retain only trajectories with correct final answers.

**Behavior Cloning.** In the initial cold-start learning phase, we distill the student policy $\hat{\pi}$ on $D_T$ via behavior cloning, minimizing $\mathcal{L}_{BC}$ as defined in (1). The resulting initialized model, $\hat{\pi}_{init}$, learns the *Thought–Code–Observation* loop. This capability enables $\hat{\pi}_{init}$ to attempt tasks without immediate failure, thus allowing fine-grained, one-step teacher interventions in subsequent phases.

## 4.2. Mentored Problem-Solving and Further SFT

After initialization via BC, the core phase, *Mentored Problem-Solving* (MPS), employs the BC-initialized student as an explorer, producing ability-matched and deficiency-localized trajectory data.

**Student Explores, Teacher Corrects.** As shown in Figure 2(a), the initialized student model $\hat{\pi}_{init}$ attempts unseen tasks. For each task, it independently generates a full trajectory $\tau_S = (\sigma_1, \dots, \sigma_H)$, where each step $\sigma_i$ denotes a triplet of *Thought–Code–Observation*. After generating the final answer, the teacher $\pi_E$ checks correctness *using the student's full reasoning chain*. If the answer is incorrect, the teacher locates the first deviation step $\sigma_k$ where the student diverges from the correct path. Instead of discarding the trajectory, $\pi_E$ provides a minimal intervention by replacing

$\sigma_k$ with a corrected step $\sigma'_k$, after which the student resumes execution from $(\sigma_1, \ldots, \sigma_{k-1}, \sigma'_k)$.

**Validation of the Correction.** If the student ultimately completes the task, it indirectly validates the teacher's correction, and the corrected trajectory is collected as training data. If the student makes another mistake at step $m > k$, the teacher will correct it to $\sigma'_m$, and the execution will continue from $(\sigma_1, \ldots, \sigma_{k-1}, \sigma'_k, \ldots, \sigma_{m-1}, \sigma'_m)$. Each intervention addresses only the specific wrong step, and a single trajectory may contain multiple such interventions. The entire process of error localization and correction is carried out by the teacher via prompts (Appendix E).

In rare cases, a task may remain unsolved even after multiple one-step corrections (5 attempts in this paper). Such cases are **Hard-to-Teach tasks**, as their difficulty exceeds the student's current capability. While these trajectories offer limited value for capability-matched SFT, we retain a subset as challenging examples for RL to improve the model's performance on difficult tasks.

**Two Complementary Forms of Supervision.** The MPS process yields two complementary forms of supervision: (1) The final *corrected trajectory*, mostly generated by the student with sparse teacher edits, provides capability-aligned demonstrations for continued SFT. (2) Each key-step correction produces a *preference pair*, anchored on the same prefix $(\sigma_1, \ldots, \sigma_{k-1})$, where the teacher's corrected step $\sigma'_k$ is preferred over the student's original $\sigma_k$. A single task can generate multiple such pairs from multiple teacher interventions. These pairs are especially useful for RL methods (e.g., GRPO (Shao et al., 2024)), offering near-correct prefixes that stabilize short rollouts and provide precise feedback at prior error points. Once this high-quality, capability-aligned data is collected, it is used to train the student model $\hat{\pi}$ via SFT (see Figure 2(b)). Training on such capability-aligned data helps reduce compounding errors; we next provide a theoretical analysis.

**Theoretical Justification.** To analyze *SCoRe*'s advantage in mitigating compounding errors (Ross et al., 2011), we model the agent as a finite-horizon process of length $H$. At each step $t$, in state $s_t \in \mathcal{S}$, the agent selects action $a_t \in \mathcal{A}$ via policy $\pi$. In our framework, $a_t$ is the composite output of thought and code. The environment transitions according to $P(s_{t+1} \mid s_t, a_t)$. Let $\pi_E$ and $\hat{\pi}$ be the teacher and student policies, and let $d_t^\pi$ be the state distribution under $\pi$. The per-step cost $c_t(s) \in [0, 1]$ is 0 if the policy acts correctly and 1 otherwise. The total expected cost is

$$c(\pi) = \mathbb{E}_{s_1 \sim d_1, \, a_t \sim \pi, \, s_{t+1} \sim P} \left[ \sum_{t=1}^{H} c_t(s_t) \right]. \quad (3)$$

**Theorem 4.1** (BC compounding-error bound). *If the student*

$\hat{\pi}$ *trained on teacher* $\pi_E$ *demonstrations via BC satisfies*

$$\mathbb{P}_{s \sim d_t^{\pi_E}} \left[ \hat{\pi}(s) \neq \pi_E(s) \right] \leq \varepsilon, \quad \forall t \in \{1, \ldots, H\}, \quad (4)$$

*then*

$$c(\hat{\pi}) \leq c(\pi_E) + \frac{H(H-1)}{2} \varepsilon = c(\pi_E) + O(H^2 \varepsilon). \quad (5)$$

This classical covariate-shift result (Ross et al., 2011) shows that a small per-step error $\varepsilon$ can grow to $O(H^2 \varepsilon)$, severely degrading performance on long-horizon tasks.

**Theorem 4.2** (*SCoRe* first-error-correction bound). *In* SCoRe *training, at the first deviation from teacher policy* $\pi_E$, *the action is replaced by the teacher's, and execution continues with student policy* $\hat{\pi}$. *If, **under** $d_t^{\hat{\pi}}$, the per-step misalignment satisfies*

$$\mathbb{P}_{s \sim d_t^{\hat{\pi}}} \left[ \hat{\pi}(s) \neq \pi_E(s) \right] \leq \varepsilon, \quad \forall t \in \{1, \ldots, H\}. \quad (6)$$

*Since* SCoRe*'s training data originates from the student's own rollouts, the relevant error rate is evaluated **under** $d_t^{\hat{\pi}}$ **rather than** $d_t^{\pi_E}$. Then*

$$c(\hat{\pi}) \leq c(\pi_E) + H \varepsilon = c(\pi_E) + O(H \varepsilon). \quad (7)$$

Unlike BC in Theorem 4.1, *SCoRe* trains on $d_t^{\hat{\pi}}$ and the teacher's first-error corrections, truncating error propagation. At most one unchecked mistake occurs before resuming an expert-aligned path, reducing worst-case growth from $O(H^2)$ to $O(H)$. The proof and a discussion of the underlying assumptions are provided in Appendix B.

### 4.3. RL Refinement for Mastery

Although MPS yields capability-aligned supervision, training the student on this data only via SFT still limits the student to replicating the teacher's key steps. To move from imitation to independent problem-solving, we introduce an RL phase after SFT that directly optimizes task success. We adopt GRPO (Shao et al., 2024), a variant of PPO (Schulman et al., 2017) that omits the value function, thereby reducing computation and value-estimation instability. However, for long-horizon tasks, GRPO remains sensitive to sparse rewards and high gradient variance. As Figure 2(c) shows, we address these issues with two techniques:

*Short-Horizon Rollout.* Instead of starting rollouts from the initial question, we begin from the verified prefix $(\sigma_1, \ldots, \sigma_{k-1})$: the sequence preceding the original error step $\sigma_k$. This shortens the horizon from $H$ to $H' = H - (k-1)$, reducing variance.

*Key-Step Reward.* If the final answer is correct, a large reward $R_{\text{final}}$ is given. Otherwise,

$$R = \begin{cases} R_{\text{key}}, & a_k = a_k^{\pi_E}, \\ R_{\text{format}}, & a_k \neq a_k^{\text{orig}} \wedge a_k \neq a_k^{\pi_E}, \\ 0, & \text{otherwise}, \end{cases}$$

where $a_k^{\mathrm{orig}}$ is the student's original error, and $a_k^{\pi_E}$ is the teacher's correction. $R_{\mathrm{format}}$ follows the format-reward design of DeepSeek-R1 (Guo et al., 2025), with a slightly stricter condition: it is granted only when the output is both format-correct and different from the original error. Action equivalence can be reliably checked via code and code execution results.

In practice, we compute the reward via a lightweight LLM-based verifier: `Qwen2.5-7B-Instruct` judges semantic equivalence between the generated and reference answers. This reward scheme provides informative credit assignment at the student's weakest step while maintaining prioritization of final task success.

As noted earlier, we retain a subset of Hard-to-Teach samples (10% of the data) specifically for RL training. The teacher provides no useful guidance on them, and the student is free to search for strategies that go beyond the teacher's behavior, rather than imitating it.

**Theoretical Justification.** We analyze variance reduction under policy gradients (ignoring value-function or clipping for clarity), aligning with GRPO's core update. The conclusion can also extend to the clipped objective of GRPO. Specifically, starting rollouts after the verified prefix reduces the remaining horizon from $H$ to $H - k + 1$, tightening the variance bound of the policy gradient:

$$g_k = \sum_{t=k}^{H} \nabla_\theta \log \pi_\theta(a_t \mid s_t) \cdot G_t, \quad G_t = \sum_{t'=t}^{H} \gamma^{t'-t} r_{t'}.$$
(8)

$\gamma \in (0,1)$ is the discount factor. Assume that the reward satisfies $|r_t| \le R_{\max}$ and $\|\nabla_\theta \log \pi_\theta(a_t \mid s_t)\| \le G_{\max}$.

**Theorem 4.3** (Variance Bound for Shortened Rollout). *Under these assumptions, there exists $C = G_{\max}^2 R_{\max}^2$ such that*

$$\mathrm{Var}[g_k] \le \frac{C}{(1-\gamma)^2} \left( (H-k+1) - \frac{\gamma(1-\gamma^{H-k+1})}{1-\gamma} \right)^2,$$
(9)

*which decreases monotonically as $k$ increases. The proof is in Appendix B.*

In summary, starting rollouts from verified prefixes focuses on students' weak points and reduces gradient variance by truncating the horizon, leading to more stable RL training.

## 5. Experiments

### 5.1. Experimental Setup

**Datasets.** We evaluate our approach on three categories of datasets. *Mathematical Reasoning*: AIME2024, AIME2025, MATH500 (Lightman et al., 2023; Hendrycks et al., 2021), and OlymMath (Sun et al., 2025). These problems often require multi-step reasoning and frequent use of a code interpreter. *Factual Reasoning*: HotpotQA (HQA), 2Wiki-MultihopQA (2Wiki) (Yang et al., 2018; Ho et al., 2020), Musique (MuSiQ) (Trivedi et al., 2022), and Bamboogle (Bamb) (Press et al., 2023). These tasks demand multi-step information retrieval and synthesis via search APIs. For *Deep Search*, *which is fully absent from the training data*, we follow WebThinker (Li et al., 2025c)'s text-only split and test on GAIA (Mialon et al., 2024), WebWalker (Wu et al., 2025), Humanity's Last Exam (HLE) (Phan et al., 2025), and xBench (Chen et al., 2025). These benchmarks involve much longer tool-use horizons, making them more challenging than standard reasoning tasks.

**Baselines.** We compare *SCoRe* with the following three categories of baselines:

(1) *Prompt-Only Large Model Agents*: strong LLMs prompted in a tool-integrated reasoning (TIR) format (Li et al., 2025b), without parameter updates. This setting reflects an approximate upper bound for performance under ideal prompting. We include `DeepSeek-V3 (671B)` and `Qwen2.5-72B-Instruct`.

(2) *Behavior Cloning* (Kang et al., 2025)[4]: the student imitates full teacher trajectories, with no exploration or correction. Note that *SCoRe* uses the same total number of training tasks as Behavior Cloning in Table 1 and 2.

(3) *Trajectory-Level RL Algorithm*: training the student with RL over the full horizon with sparse task-completion rewards, using the GRPO (Shao et al., 2024) or ARPO (Dong et al., 2026) algorithm. Note that RL baselines are also warm-started with BC, rather than trained from scratch.

We also evaluate stronger teacher-guided baselines in Appendix D, including a PRM-style step-reward baseline, on-policy distillation, and ReLIFT (Ma et al., 2026); these baselines use richer supervision and higher training cost.

Our proposed *SCoRe* is evaluated in two sequential phases: (i) *SCoRe-SFT*, a correction-based SFT phase where the model is trained on capability-matched and deficiency-localized data; and (ii) *SCoRe-RL*, a short-horizon RL phase with rollouts from the verified prefix and key-step rewards for reproducing the teacher's fix or avoiding errors.

**Evaluation.** To match the evaluation settings of ARPO (Dong et al., 2026), we measure open-domain QA tasks using a token-level F1 score against the ground truth. For mathematical reasoning and deep search tasks, correctness is judged by `Qwen2.5-72B-Instruct` by compar-

---

[4]Our *SCoRe* framework also includes a BC phase for cold-start initialization, but unlike the BC baseline in Table 1 and 2, which uses the full teacher-annotated training set, the BC phase of *SCoRe* uses only 20% of the data to endow the student with basic reasoning–action skills.

*Table 1.* Overall performance on eight challenging reasoning tasks for Student: Qwen2.5-7B-Instruct. The best results among same-size models are bolded. "Avg." denotes the average across tasks. Results for GRPO and ARPO are mostly from Dong et al. (2026), and all evaluation settings follow Dong et al. (2026). **More base models and distillation baselines are reported in Appendix Tables 6 and 7.** Since most reasoning benchmarks were released before 2024, we do not use Qwen3-8B (training data cutoff to October 2024) due to potential data leakage risks. AIME24 and AIME25 results are averaged over five runs due to their small size.

| Method | Mathematical Reasoning | | | | Factual Reasoning | | | | Avg. |
|---|---|---|---|---|---|---|---|---|---|
| | AIME24 | AIME25 | MATH500 | OlymMath | HQA | 2Wiki | MuSiQ | Bamb | |
| **Larger Models (TIR Prompting)** | | | | | | | | | |
| DeepSeek-V3 (671B) | 44.0 | 33.3 | 84.8 | 21.0 | 60.1 | 78.5 | 37.0 | 73.2 | 54.0 |
| Qwen2.5-72B-Instruct | 32.7 | 35.3 | 77.4 | 17.0 | 60.5 | 75.5 | 36.8 | 73.2 | 51.1 |
| **Student: Qwen2.5-7B-Instruct** | | | | | | | | | |
| Behavior Cloning (BC) | 22.0 | 14.7 | 72.8 | 15.5 | 58.1 | 70.3 | 26.6 | 63.6 | 43.0 |
| BC + GRPO | 22.6 | 26.0 | 78.0 | 25.0 | 59.0 | 76.1 | 30.6 | 68.4 | 48.2 |
| BC + ARPO | 32.0 | 27.4 | 78.8 | 18.0 | 58.8 | 76.1 | 31.1 | 71.5 | 49.2 |
| SCoRe-SFT | 26.7 | 18.7 | 73.4 | 18.5 | 59.5 | 72.8 | 29.2 | 69.8 | 46.1 |
| SCoRe-RL | **38.0** | **28.7** | **82.0** | **26.5** | **61.4** | **76.8** | **32.2** | **72.8** | **52.3** |

*Table 2.* Performance on more free-form benchmarks for **deep search**. Given their difficulty and need for stronger base reasoning, we include the latest Qwen3-8B for evaluation. Since most deep search benchmarks were released in 2025, this poses little risk of data leakage. More baselines (PRM-style, On-Policy Distillation, ReLIFT (Ma et al., 2026)) are presented in Table 8 of Appendix D.

| Method | GAIA | HLE | XBench | WebWalker | Avg. |
|---|---|---|---|---|---|
| **Larger Models (TIR Prompting)** | | | | | |
| DeepSeek-V3 (671B) | 40.8 | 10.6 | 32.0 | 46.0 | 32.4 |
| Qwen2.5-72B-Instruct | 32.0 | 7.8 | 31.0 | 38.5 | 27.3 |
| **Student: Qwen3-8B (Non-thinking)** | | | | | |
| Behavior Cloning (BC) | 27.2 | 8.4 | 21.0 | 34.5 | 22.8 |
| BC + GRPO | 32.0 | 7.8 | 20.0 | 29.0 | 22.2 |
| BC + ARPO | 38.8 | 8.8 | 25.0 | 30.5 | 25.8 |
| SCoRe-SFT | 28.2 | 10.0 | 22.0 | 41.5 | 25.4 |
| SCoRe-RL | **40.8** | **11.0** | **27.0** | **43.0** | **30.5** |

ing the generated final answer with the provided ground truth. In Table 7 of Appendix D, **we also report Exact Match (EM)** for a more objective evaluation.

**Implementation.** During training and inference, the model employs two tools: a Python code interpreter for mathematical reasoning and precise computation, and an online search engine providing concise, up-to-date snippets. To save time and cost, *we did not use a web browser in deep search tasks.* SFT training data follows the [Thought-Action-Observation] format with Qwen2.5-72B-Instruct as the teacher. To construct the training data, we collect seed question–answer pairs mainly from the Tool-Star dataset (Dong et al., 2025), including NuminaMath (Li et al., 2024) and Omni (Gao et al., 2025) for math reasoning, and HotpotQA, 2Wiki-MultiHopQA, and WebWalker for factual QA. Of these, 20% of the trajectories are fully teacher-annotated to create high-quality BC initialization data, giving the student a solid foundation in reasoning–acting skills. The remaining 80% are generated via MPS, producing capability-matched,

deficiency-localized data. These minimally corrected trajectories are split evenly: half for correction-based SFT to produce the *SCoRe-SFT* model, and half for RL. Note that during MPS, a substantial portion of the data can be solved correctly by the student without any teacher intervention. We retain only a small subset of these easy cases for training.

During RL training, the maximum number of rollout steps was set to 8, and any reasoning–action cycles exceeding this limit were considered incomplete. The same configuration was used during inference. For the key-step reward scale $R_{\text{key}}$, we set it to 0.5 based on a small-scale 2k-sample search over $\{0.3, 0.5, 0.7\}$ on math benchmarks, where 0.5 achieved the best performance. For the format reward $R_{\text{format}}$, we use the standard value of 0.1. Additional details are provided in Appendix C.

### 5.2. Main Results

**TIR prompting excels on ultra-large models but is costly.** On reasoning tasks, Tool-Integrated Reasoning (TIR) is ef-

*Table 3.* Ablation study of various components of *SCoRe*. Student: `Qwen2.5-7B-Instruct`.

| Phase | Design | Mathematical Reasoning | | | | Factual Reasoning | | | | Avg. |
|---|---|---|---|---|---|---|---|---|---|---|
| | | AIME24 | AIME25 | MATH500 | OlymMath | HQA | 2Wiki | MuSiQ | Bamb | |
| SFT | Initial Distillation (BC init) | 20.0 | 23.3 | 70.8 | 7.0 | 54.4 | 67.6 | 27.0 | 63.7 | 41.7 |
| SFT | SFT on MPS data (SCoRe-SFT) | 26.7 | 18.7 | 73.4 | 18.5 | 59.5 | 72.8 | 29.2 | 69.8 | 46.1 |
| RL | RL W/O short-horizon | 30.0 | 20.0 | 78.0 | 25.0 | 58.0 | 75.3 | 30.2 | 70.7 | 48.4 |
| RL | RL W/O key-step rewards | 33.3 | 20.0 | 80.6 | 24.0 | 61.1 | 76.7 | 30.3 | 71.7 | 49.7 |
| RL | Full design (SCoRe-RL) | **38.0** | **28.7** | **82.0** | **26.5** | **61.4** | **76.8** | **32.2** | **72.8** | **52.3** |

fective for very large models: `DeepSeek-V3 (671B)` scores 54.0 on average, and `Qwen2.5-72B-Instruct` scores 51.1, showing how TIR combines strong LLM reasoning with the precision of external tools. But in agent settings, such models also incur high latency and token costs, which grow with both token volume and interaction turns.

**SCoRe matches teacher performance on reasoning and deep search with smaller backbones.** Tables 1 and 2 report results on reasoning and deep search. On smaller backbones, our method uses MPS trajectories from a student-led process with one-step teacher correction, and trains the model with correction-based SFT and short-horizon RL with key-step rewards. This yields large gains: averaged over math and factual reasoning benchmarks, *SCoRe-RL* with `Qwen2.5-7B-Instruct` scores 52.3, and outperforms BC by 9.3 and GRPO by 4.1. Similar trends hold for other base models (`Qwen2.5-3B-Instruct` and `Llama-3.1-8B-Instruct`; Appendix Table 6). Ability-aligned training thus enables small models to match much larger ones at lower cost.

**SCoRe-SFT outperforms behavior cloning under the same data budget.** With only teacher-annotated trajectories, BC performance is limited. Under the same data budget, *SCoRe-SFT* yields consistent gains across backbones on both reasoning and deep search, showing that MPS-constructed data reduces compounding error growth from $O(H^2)$ to $O(H)$, where $H$ is the horizon length. This also shows that even without RL, training on correction-based trajectories substantially outperforms BC distillation.

**RL with short-horizon and key-step rewards drives strong performance.** On the more challenging Deep Search tasks (Table 2), *SCoRe-RL* achieves very strong performance and in some cases exceeds the teacher. For `Qwen3-8B` (Non-thinking mode), it scores 30.5 (+7.7 over BC, +8.3 over GRPO). This is expected: unlike imitation learning, *the learning signal in RL comes not only from the teacher but also from the environment*. Since the teacher provides only local corrections rather than an optimal policy, *SCoRe-RL* can explore beyond the teacher's strategies, achieving a large reward $R_{\text{final}}$ (as described in Section 4.3).

Although the RL baseline is competitive, *SCoRe-RL* remains

better. Moreover, this advantage is not simply due to extra teacher guidance or compute: in Appendix Table 8, *SCoRe-RL* also outperforms stronger teacher-guided baselines such as PRM-style step rewards and on-policy distillation.

### 5.3. Ablation Study

In the ablation study, we evaluate the contribution of each *SCoRe* component on reasoning benchmarks (Table 3). The BC-initialized student (Initial Distillation), trained on a subset of high-quality teacher trajectory data, serves as the explorer for MPS. Unlike the BC baselines in Tables 1 and 2, which use the full teacher data, this model performs poorly. In contrast, when the initial model is treated as the explorer and fine-tuned on data generated via MPS, performance improves substantially (*SCoRe-SFT*), showing that training on such data effectively reinforces weak links in the reasoning chain. Extending to RL brings further improvements: (1) *Without short-horizon rollouts*, stability degrades and performance drops notably, showing that truncation reduces gradient-estimation variance; (2) *Without key-step rewards*, accuracy on multi-step tasks decreases, validating the need for targeted, stepwise guidance beyond final-task rewards.

The full *SCoRe-RL* approach, combining both short-horizon rollouts and key-step rewards, achieves the best results.

Although the full *SCoRe* pipeline requires extra compute due to MPS, the ablations in Table 3 are compute-matched, using the same MPS-generated data and the same GRPO training pipeline as ours. Hence, the gains of the full design are not simply a byproduct of higher training cost.

### 5.4. RL Helps Solve Previously Hard-to-Teach Cases

We further analyze teacher–student interaction during MPS (Figure 3(a)). Most tasks require only a single teacher correction, while very few need $\geq 2$ interventions, suggesting that one-step correction is usually sufficient. Cases where the student still fails after several one-step corrections are labeled as *Hard-to-Teach*. Although these cases receive no teacher guidance and make up a small fraction of the data, we retain a subset for RL training rather than discard them.

This design is also cost-efficient. The average token usage

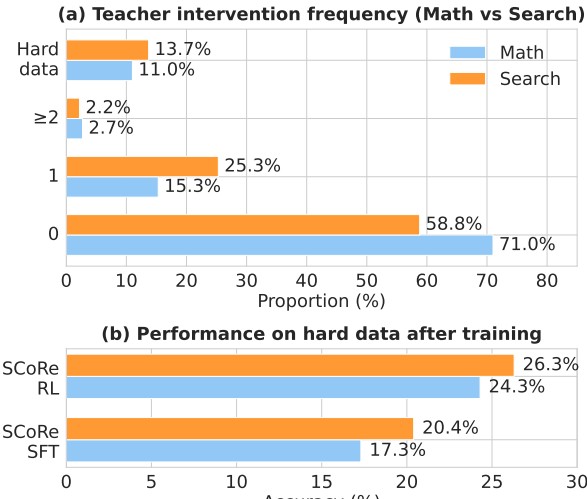

**(a) Teacher intervention frequency (Math vs Search)**

**(b) Performance on hard data after training**

*Figure 3.* Teacher intervention frequency during the MPS phase, and performance on Hard-to-Teach data after training. Categories: 0 = solved by student alone, 1 = 1 teacher correction, $\geq 2$ = 2 or more corrections. Hard data = unsolved cases, even with the teacher's help.

*Table 4.* A representative Hard-to-Teach case that becomes solvable after RL.

| |
|---|
| **Question:** What role is the actor who starred with Glynn Turman in *J.D.'s Revenge* best known for? **Labeled Answer:** Gunnery Sergeant Emil Foley |
| **Before RL:** Identify actors in *J.D.'s Revenge* → retrieve a plausible co-star (e.g., *Louis Gossett Jr.*) → retrieve multiple well-known roles for that actor → return the first plausible one without further checking. **After RL:** Identify actors in *J.D.'s Revenge* → retrieve *Louis Gossett Jr.* as the relevant co-star of Glynn Turman → observe multiple plausible notable roles → issue an extra verification query for *Gunnery Sergeant Emil Foley* → use the verified result as the final answer. **Observed change:** RL adds an explicit verification step, rather than merely reproducing teacher corrections. |

per teacher intervention, including both prompt and output tokens, is 811 for math tasks and 1260 for search tasks when only one intervention is needed, rising to 1275 and 2046, respectively, when two interventions are required.

As shown in Figure 3(b), on a held-out Hard-to-Teach subset (200 items excluded from training), SFT on MPS data raises performance from 0% to 17.3% on math and 20.4% on search, while RL further improves it to 24.3% and 26.3%, respectively. This suggests that Hard-to-Teach during MPS does not mean unlearnable: on-policy RL can further improve the policy and solve some previously unsolved cases.

To better understand these gains, we manually inspected 50 cases that were Hard-to-Teach under MPS but became solvable after RL. Only 12% were best explained by generation instability, such as a previously malformed tool call being formatted correctly after RL. In most cases, the improvement reflected interpretable strategy refinement, including better decomposition, retrieval, or added verification. Table 4 shows one example: RL prompts the model to verify before submitting the answer, avoiding its earlier mistake. These findings suggest that the gains stem primarily from interpretable strategy learning rather than randomness. More broadly, RL's on-policy nature allows the student's agentic capabilities to improve gradually over training, rather than solving Hard-to-Teach cases through sudden "epiphanies."

## 6. Conclusion

We introduce *SCoRe*, a framework for distilling LLM agents that lets students actively explore problem-solving with min-

imal, targeted teacher corrections. This yields training data aligned with the student's evolving capabilities and reveals weaknesses for further optimization, helping students advance from imitation to genuine problem-solving. Extensive experiments show that *SCoRe* achieves expert-level performance and consistently outperforms standard distillation baselines. Future work includes improving reward design and extending the framework to broader multimodal tasks.

## Limitations

Despite reducing compounding error, the method does not eliminate it. The $O(H)$ result should therefore be interpreted as a conditional theoretical characterization under the stated assumptions, rather than as a guarantee that distribution shift is fully resolved in practice. The framework also depends on teacher quality during MPS. A weaker teacher may reduce the amount of usable training data, but the validation mechanism typically filters out incorrect corrections, so its effect is more directly on data yield than on supervision quality.

## Impact Statement

This paper presents work whose goal is to advance the field of machine learning. There are many potential societal consequences of our work, none of which we feel should be specifically highlighted here.

## Acknowledgments

This work was supported in part by the grants from National Science and Technology Major Project (No. 2023ZD0121104), and the Anhui Natural Science Foundation (No. 2508085ZD006).

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

## The Use of Large Language Models (LLMs) in Writing

An LLM (i.e., OpenAI's GPT-5) was used solely for minor language editing, including grammar correction and slight rephrasing for clarity. It did not contribute to the research design, and all scientific content is entirely the authors' own.

## A. Relation to Closely Related Work

**Why traditional DAgger cannot be directly used in agent distillation:** DAgger (Ross et al., 2011) was designed in **2011** for classical imitation-learning domains (e.g., robotic control), where (i) the state is well-defined and can be judged locally, and (ii) expert labels are treated as reliable supervision. These assumptions break down for LLM agents that act through long-horizon, partially observable, tool-augmented interaction.

Our *SCoRe* method evaluates the entire trajectory and identifies mistakes only after seeing the global trajectory, which is necessary to correctly pinpoint where errors occurred. In contrast, DAgger forces the teacher to correct the agent at every step, even though many actions reveal themselves as right or wrong only later through tool feedback or final success or failure. This is suitable for traditional imitation learning but not for agent distillation. Moreover, the teacher model itself can make mistakes or contain noise; DAgger fully trusts the teacher, whereas *SCoRe* automatically filters out teacher errors, which is crucial.

**Other Closely Related Work:** *SCoRe* is related to prior correction-based methods such as LEMMA (Pan et al., 2025), which identify an erroneous step and regenerate from that point. Its key distinction is that *SCoRe* remains student-centered, which is also the main focus of this paper: after correction, the student itself continues the trajectory, and the resulting verified prefix is further used for short-horizon RL. This strict student-centered design is required by our theory.

*SCoRe* is also related to PRM-style methods with step-level rewards (Zhang et al., 2025), but it does not train a separate process reward model. Instead, it uses first-error localization on student-generated trajectories to define both rollout starting points and localized rewards.

## B. Proofs of Theoretical Results

**Remark on assumptions.** Our analysis adopts the same standard finite-horizon assumptions used in the DAgger compounding-error analysis (Ross et al., 2011), including bounded per-step costs and per-step policy deviation probabilities under the relevant state distribution. These assumptions are not introduced specifically for *SCoRe*; they are the same classical assumptions under which the widely adopted $O(H^2)$ compounding-error bound for behavior cloning is derived. Our $O(H)$ result should be interpreted as a comparative bound within this established framework, rather than as an unconditional guarantee for all practical LLM-agent settings.

**Theorem B.1** (BC compounding-error bound). *Suppose the student policy $\hat{\pi}$ has a one-step deviation probability of at most $\varepsilon$ under the expert's state distribution, i.e.,*

$$\mathbb{P}_{s \sim d_t^{\pi_E}} \left( \hat{\pi}(s) \neq \pi_E(s) \right) \leq \varepsilon,$$

*and that per-step costs are bounded as $c_t \in [0, 1]$. Then the student policy satisfies*

$$c(\hat{\pi}) \leq c(\pi_E) + \frac{H(H-1)}{2} \varepsilon = c(\pi_E) + O(H^2 \varepsilon). \tag{10}$$

The proof below follows the approach of (Ross et al., 2011).

*Proof.* The core of this proof is to quantify how errors accumulate over time. A single mistake can lead the student policy into a state distribution the teacher has never seen, causing further deviations. We will bound the total cost by analyzing the probability of such deviations at each step.

For any step $t \in \{1, \ldots, H\}$, define the event $\mathcal{E}_{t-1}$ as the student making no errors in the first $t-1$ steps, assuming it encounters states from the expert's distribution $d_i^{\pi_E}$:

$$\mathcal{E}_{t-1} := \bigcap_{i=1}^{t-1} \{\hat{\pi}(s_i) = \pi_E(s_i)\}, \quad \text{where } s_i \sim d_i^{\pi_E}. \tag{11}$$

The complementary event, $\mathcal{E}_{t-1}^c$, represents at least one error occurring before step $t$. By the union bound:

$$\mathbb{P}(\mathcal{E}_{t-1}^c) = \mathbb{P}\left(\bigcup_{i=1}^{t-1}\{\hat{\pi}(s_i) \neq \pi_E(s_i)\}\right) \leq \sum_{i=1}^{t-1}\mathbb{P}(\hat{\pi}(s_i) \neq \pi_E(s_i)) \leq (t-1)\varepsilon. \tag{12}$$

The student's state distribution at step $t$, $d_t^{\hat{\pi}}$, depends on whether an error occurred previously:

- If $\mathcal{E}_{t-1}$ occurs (no prior errors), then $d_t^{\hat{\pi}} = d_t^{\pi_E}$.

- If $\mathcal{E}_{t-1}^c$ occurs (an error was made), the student's trajectory diverges, leading to an arbitrary worst-case state distribution, denoted $q_t$.

By the law of total probability, we can express $d_t^{\hat{\pi}}$ as a mixture:

$$d_t^{\hat{\pi}} = (1 - p_{t-1})\, d_t^{\pi_E} + p_{t-1}\, q_t, \tag{13}$$

where $p_{t-1} := \mathbb{P}(\mathcal{E}_{t-1}^c)$.

We can now bound the expected cost for the student at step $t$:

$$\begin{aligned}
\mathbb{E}_{s\sim d_t^{\hat{\pi}}}[c_t(s)] &= (1 - p_{t-1})\,\mathbb{E}_{s\sim d_t^{\pi_E}}[c_t(s)] + p_{t-1}\,\mathbb{E}_{s\sim q_t}[c_t(s)] \\
&\leq (1 - p_{t-1})\,\mathbb{E}_{s\sim d_t^{\pi_E}}[c_t(s)] + p_{t-1}\cdot 1 \quad (\text{as } c_t \in [0,1]) \\
&= \mathbb{E}_{s\sim d_t^{\pi_E}}[c_t(s)] - p_{t-1}\,\mathbb{E}_{s\sim d_t^{\pi_E}}[c_t(s)] + p_{t-1} \\
&\leq \mathbb{E}_{s\sim d_t^{\pi_E}}[c_t(s)] + p_{t-1} \\
&\leq \mathbb{E}_{s\sim d_t^{\pi_E}}[c_t(s)] + (t-1)\varepsilon. 
\end{aligned} \tag{14}$$

This shows the difference in expected cost at step $t$ is bounded by the cumulative error probability up to that point.

Summing over all $H$ steps:

$$\begin{aligned}
c(\hat{\pi}) - c(\pi_E) &= \sum_{t=1}^{H}\left(\mathbb{E}_{s\sim d_t^{\hat{\pi}}}[c_t(s)] - \mathbb{E}_{s\sim d_t^{\pi_E}}[c_t(s)]\right) \\
&\leq \sum_{t=1}^{H}(t-1)\varepsilon \\
&= \varepsilon\sum_{k=0}^{H-1}k = \varepsilon\cdot\frac{(H-1)H}{2}. 
\end{aligned} \tag{15}$$

Rearranging yields the result. $\qquad\square$

**Theorem B.2** (*SCoRe* first-error-correction bound). *Let $\hat{\pi}$ be a student policy trained via* SCoRe *with first-error correction, and let $\pi_E$ be the teacher policy. Assume that under the student's own induced state distribution $d_t^{\hat{\pi}}$, the per-step error rate is bounded by $\varepsilon \in [0,1]$:*

$$\mathbb{P}_{s\sim d_t^{\hat{\pi}}}[\hat{\pi}(s) \neq \pi_E(s)] \leq \varepsilon, \quad \forall t \in \{1,\dots,H\}. \tag{16}$$

*Then for any per-step cost function $c_t : \mathcal{S} \to [0,1]$, the student policy's expected total cost is bounded by*

$$c(\hat{\pi}) \leq c(\pi_E) + H\varepsilon. \tag{17}$$

*Proof.* The key advantage of the *SCoRe* framework is that the student is trained and evaluated on its own state distribution $d_t^{\hat{\pi}}$. This on-policy training avoids the covariate shift issue in standard Behavior Cloning. The proof uses a direct bound on the per-step cost difference by leveraging the on-policy error assumption.

Consider the expected cost of $\hat{\pi}$ at any step $t$. Let

$$\mathcal{G}_t := \{\hat{\pi}(s) = \pi_E(s)\}, \quad s \sim d_t^{\hat{\pi}},$$

i.e., the student takes the same action as the teacher. Its complement $\mathcal{G}_t^c := \{\hat{\pi}(s) \neq \pi_E(s)\}$ is the error event.

By the law of total expectation,

$$\mathbb{E}_{s \sim d_t^{\hat{\pi}}}[c_t(s)] = \mathbb{E}[c_t(s) \mid \mathcal{G}_t] \cdot \mathbb{P}(\mathcal{G}_t) + \mathbb{E}[c_t(s) \mid \mathcal{G}_t^c] \cdot \mathbb{P}(\mathcal{G}_t^c). \tag{18}$$

From the assumption,

$$\mathbb{P}(\mathcal{G}_t^c) \leq \varepsilon, \quad \text{so} \quad \mathbb{P}(\mathcal{G}_t) \geq 1 - \varepsilon. \tag{19}$$

Now, we bound the conditional expectations:

- If $\mathcal{G}_t^c$ occurs, the cost is at most 1 since $c_t \in [0, 1]$. Thus, $\mathbb{E}[c_t(s) \mid \mathcal{G}_t^c] \leq 1$.

- If $\mathcal{G}_t$ occurs, the student follows the teacher's action. The cost over these states is aligned with the teacher's cost. Hence,

$$\mathbb{E}[c_t(s) \mid \mathcal{G}_t] \leq \mathbb{E}_{s \sim d_t^{\pi_E}}[c_t(s)].$$

Plugging these bounds in:

$$\begin{aligned} \mathbb{E}_{s \sim d_t^{\hat{\pi}}}[c_t(s)] &\leq \mathbb{E}_{s \sim d_t^{\pi_E}}[c_t(s)] \cdot \mathbb{P}(\mathcal{G}_t) + 1 \cdot \mathbb{P}(\mathcal{G}_t^c) \\ &\leq \mathbb{E}_{s \sim d_t^{\pi_E}}[c_t(s)](1 - \mathbb{P}(\mathcal{G}_t^c)) + \mathbb{P}(\mathcal{G}_t^c) \\ &= \mathbb{E}_{s \sim d_t^{\pi_E}}[c_t(s)] - \mathbb{P}(\mathcal{G}_t^c) \cdot \mathbb{E}_{s \sim d_t^{\pi_E}}[c_t(s)] + \mathbb{P}(\mathcal{G}_t^c). \end{aligned} \tag{20}$$

As $\mathbb{E}_{s \sim d_t^{\pi_E}}[c_t(s)] \geq 0$, the negative term can be dropped:

$$\mathbb{E}_{s \sim d_t^{\hat{\pi}}}[c_t(s)] \leq \mathbb{E}_{s \sim d_t^{\pi_E}}[c_t(s)] + \mathbb{P}(\mathcal{G}_t^c). \tag{21}$$

Applying the error bound $\mathbb{P}(\mathcal{G}_t^c) \leq \varepsilon$, we get the per-step inequality:

$$\mathbb{E}_{s \sim d_t^{\hat{\pi}}}[c_t(s)] \leq \mathbb{E}_{s \sim d_t^{\pi_E}}[c_t(s)] + \varepsilon. \tag{22}$$

Summing over the horizon:

$$\begin{aligned} c(\hat{\pi}) - c(\pi_E) &= \sum_{t=1}^{H} \left( \mathbb{E}_{s \sim d_t^{\hat{\pi}}}[c_t(s)] - \mathbb{E}_{s \sim d_t^{\pi_E}}[c_t(s)] \right) \\ &\leq \sum_{t=1}^{H} \varepsilon = H\varepsilon. \end{aligned} \tag{23}$$

This yields the desired bound. $\qquad\square$

**Theorem B.3** (Variance bound for shortened rollout). *Under bounded rewards $|r_t| \leq R_{\max}$, bounded policy score norms $\|\nabla_\theta \log \pi_\theta(a_t|s_t)\| \leq G_{\max}$, and discount $\gamma \in (0, 1)$, the truncated policy gradient estimator*

$$g_k = \sum_{t=k}^{H} \nabla_\theta \log \pi_\theta(a_t|s_t) \cdot \left( \sum_{t'=t}^{H} \gamma^{t'-t} r_{t'} \right) \tag{24}$$

*satisfies*

$$\mathrm{Var}[g_k] \leq \frac{C}{(1-\gamma)^2} \left( (H - k + 1) - \frac{\gamma(1 - \gamma^{H-k+1})}{1 - \gamma} \right)^2, \quad C := G_{\max}^2 R_{\max}^2, \tag{25}$$

*and the bound decreases monotonically as $k$ increases.*

*Proof.* The estimator (24) can be written as a sum of random variables:

$$g_k = \sum_{t=k}^{H} X_t, \quad X_t := \nabla_\theta \log \pi_\theta(a_t|s_t) \, G_t, \tag{26}$$

where $G_t$ is the discounted return from $t$:

$$G_t := \sum_{t'=t}^{H} \gamma^{t'-t} r_{t'}. \tag{27}$$

Under the boundedness assumptions,

$$\|\nabla_\theta \log \pi_\theta(a_t|s_t)\| \leq G_{\max}, \tag{28}$$

$$|G_t| \leq R_{\max} \cdot \frac{1 - \gamma^{H-t+1}}{1 - \gamma}. \tag{29}$$

The norm of each term $\|X_t\|$ is therefore bounded by:

$$\|X_t\| \leq G_{\max}|G_t| \leq G_{\max} R_{\max} \frac{1 - \gamma^{H-t+1}}{1 - \gamma}. \tag{30}$$

To bound the variance, we bound the norm of the total gradient estimator $g_k$ using the triangle inequality and the time-dependent bound on $|G_t|$ from (29):

$$\|g_k\| = \left\| \sum_{t=k}^{H} X_t \right\| \leq \sum_{t=k}^{H} \|X_t\|$$

$$\leq \sum_{t=k}^{H} G_{\max} R_{\max} \frac{1 - \gamma^{H-t+1}}{1 - \gamma}$$

$$= \frac{G_{\max} R_{\max}}{1 - \gamma} \sum_{t=k}^{H} (1 - \gamma^{H-t+1}). \tag{31}$$

The summation can be computed by letting $j = H - t + 1$:

$$\sum_{j=1}^{H-k+1} (1 - \gamma^j) = (H - k + 1) - \sum_{j=1}^{H-k+1} \gamma^j = (H - k + 1) - \gamma \frac{1 - \gamma^{H-k+1}}{1 - \gamma}. \tag{32}$$

Substituting this back gives the bound on the norm of $g_k$:

$$\|g_k\| \leq \frac{G_{\max} R_{\max}}{1 - \gamma} \left( (H - k + 1) - \frac{\gamma(1 - \gamma^{H-k+1})}{1 - \gamma} \right). \tag{33}$$

The variance is bounded by the second moment, $\mathrm{Var}[g_k] \leq \mathbb{E}[\|g_k\|^2]$. Since our bound on $\|g_k\|$ is a deterministic constant, we have:

$$\mathrm{Var}[g_k] \leq \frac{C}{(1 - \gamma)^2} \left( (H - k + 1) - \frac{\gamma(1 - \gamma^{H-k+1})}{1 - \gamma} \right)^2, \tag{34}$$

where $C := G_{\max}^2 R_{\max}^2$, this bound decreases monotonically as $k$ increases. $\square$

## C. Datasets and implementation details

**Benchmarks and Evaluation Protocol.** We evaluate on exactly the same benchmark datasets and official test splits as ARPO (Dong et al., 2026), ensuring a fair comparison. Table 5 shows the composition and scale of the test benchmark. Open-domain QA tasks are measured using averaged token-level F1 against the ground truth, following ARPO's protocol,

*Table 5.* Overview of evaluation benchmarks used in our experiments.

| Category | Dataset | Test Size | Description |
|---|---|---|---|
| Math | AIME24[1] | 30 problems | 2024 AIME math problems in algebra and geometry; used to assess advanced reasoning. |
| | AIME25[2] | 30 problems | 2025 AIME I&II covering algebra, combinatorics, geometry, and number theory. |
| | MATH500 (Lightman et al., 2023) | 500 problems | High-difficulty MATH subset with university-level algebra, calculus, and number theory. |
| | OlymMath (Sun et al., 2025) | 200 problems | Olympiad-level mathematics to evaluate the model's abilities in hard cases. |
| Factual | HotPotQA (Yang et al., 2018) | 200 QA pairs | Wikipedia-based multi-hop QA on complex retrieval and reasoning. |
| | 2Wiki (Ho et al., 2020) | 200 QA pairs | Multi-document QA requiring multi-step reasoning from two Wikipedia articles. |
| | Musique (Trivedi et al., 2022) | 200 QA pairs | Multi-hop QA benchmark for semantic understanding and logical inference. |
| | Bamboogle (Press et al., 2023) | 125 QA pairs | Two-hop questions that are challenging for common web search engines yet have evidence available on Wikipedia. |
| Deep Search | GAIA (Mialon et al., 2024) | 103 queries | Reasoning, web navigation, and tool-use tasks for AI assistant evaluation. |
| | HLE (Phan et al., 2025) | 500 queries | Interdisciplinary and abstract problems demanding advanced reasoning. |
| | WebWalker (Wu et al., 2025) | 200 queries | Web-navigation tasks in dynamic, multi-hop retrieval settings. |
| | xBench (Chen et al., 2025) | 100 queries | Deep-search test of breadth and depth in agent reasoning. |

[1] `https://huggingface.co/datasets/HuggingFaceH4/aime_2024`
[2] `https://huggingface.co/datasets/math-ai/aime25`

while correctness on mathematical reasoning and deep search tasks is judged by `Qwen2.5-72B-Instruct` under the *LLM-as-a-judge* paradigm (Zheng et al., 2023). In Appendix Table 7, **we also report Exact Match (EM)**.

**Supervised Fine-Tuning.** For BC initialization and *SCoRe-SFT*, we fine-tune all backbones using the LLaMAFactory framework (Zheng et al., 2024) with a learning rate of $7 \times 10^{-6}$, the AdamW optimizer with weight decay 0.1, and a global batch size of 128 for 3 epochs. Training employs DeepSpeed ZeRO-3 (Rasley et al., 2020) and FlashAttention-2 (Dao, 2024) for efficiency, with BF16 mixed precision and a maximum sequence length of 4096 tokens. All tool execution outputs are excluded from the loss; we compute the loss only on natural-language reasoning text and tool invocation requests, ensuring that the model focuses on reasoning quality rather than memorizing tool responses. The training data consists of two distinct categories: search and math. For BC initialization, we use 2,031 search trajectories and 2,080 math trajectories, while *SCoRe-SFT* employs 4,990 search and 5,019 math trajectories for correction-based fine-tuning.

**Reinforcement Learning.** The *SCoRe-RL* phase performs short-horizon fine-tuning using the GRPO algorithm (Shao et al., 2024) within the VERL framework (Sheng et al., 2025). We adopt a global batch size of 128, a PPO mini-batch size of 16, a rollout size of 16, and a maximum response length of 4,096 tokens. Training is conducted for 3 epochs on 8×NVIDIA H20 GPUs, using 5,271 search trajectories and 5,639 math trajectories.

For factual reasoning datasets, the search tool is implemented via the Google Search API. To reduce tool-call latency during both training and inference, we include only the textual snippets returned by the API as the [Observation] content, omitting full browser navigation or long-form summarization. This design substantially reduces retrieval time while preserving sufficient context to answer queries effectively.

*Table 6.* Overall performance on eight challenging reasoning tasks for other base models. The best outcomes among models of the same size are **bolded**. "Avg." denotes the average score across tasks. Results for GRPO and ARPO are mostly taken from (Dong et al., 2026).

| Method | Mathematical Reasoning | | | | Factual Reasoning | | | | Avg. |
|---|---|---|---|---|---|---|---|---|---|
| | AIME24 | AIME25 | MATH500 | OlymMath | HQA | 2Wiki | MuSiQ | Bamb | |
| **Student: Qwen2.5-3B-Instruct** | | | | | | | | | |
| Behavior Cloning (BC) | 13.3 | 13.3 | 65.6 | 9.5 | 51.7 | 63.7 | 26.8 | 61.2 | 38.3 |
| BC + GRPO | 20.0 | 13.3 | 72.0 | 16.0 | 56.5 | 64.5 | 24.7 | 65.2 | 41.5 |
| BC + ARPO | 20.0 | 20.0 | 71.4 | 14.5 | 58.5 | 67.4 | 28.7 | 66.8 | 43.4 |
| SCoRe-SFT | 20.0 | 13.3 | 67.0 | 12.5 | 55.9 | 71.6 | 27.8 | 67.5 | 42.0 |
| SCoRe-RL | **26.7** | **20.0** | **72.4** | **17.5** | **59.2** | **75.6** | **29.0** | **73.2** | **46.7** |
| **Student: Llama3.1-8B-Instruct** | | | | | | | | | |
| Behavior Cloning (BC) | 6.7 | 6.7 | 58.0 | 8.5 | 55.6 | 65.5 | 28.6 | 69.1 | 37.3 |
| BC + GRPO | 13.3 | 13.3 | 62.4 | 14.5 | 57.8 | 71.8 | 31.0 | 68.2 | 41.5 |
| BC + ARPO | 23.3 | 16.7 | 64.6 | 11.0 | **65.4** | 75.5 | 34.8 | 73.8 | 45.6 |
| SCoRe-SFT | 6.7 | 10.0 | 60.0 | 10.0 | 59.5 | 71.7 | 31.1 | 71.4 | 40.1 |
| SCoRe-RL | **26.7** | **20.0** | **65.4** | **16.0** | 64.2 | **76.5** | **35.3** | **76.2** | **47.5** |

*Table 7.* Comparison with Agent Distillation (Kang et al., 2025) on mathematical tasks. Results with a gray background are reported from the original paper. All math tasks are evaluated by exact match, consistent with Kang et al. (2025).

| Method | MATH500 | GSM-Hard | AIME | OlymMATH | Avg. |
|---|---|---|---|---|---|
| **Qwen2.5-32B-Instruct** | | | | | |
| CoT Prompting | 79.2 | 74.6 | 13.3 | 6.0 | 43.3 |
| Agent Prompting | 69.2 | 76.4 | 21.1 | 11.5 | 44.6 |
| **Student: Qwen2.5-7B-Instruct** | | | | | |
| CoT Distillation + RAG | 68.0 | 60.6 | 6.7 | 5.0 | 35.1 |
| Agent Distillation | 67.8 | 72.4 | 15.6 | 11.5 | 41.8 |
| *SCoRe-RL* | 80.2 | 79.7 | 21.2 | 16.0 | 49.3 |
| **Student: Qwen2.5-3B-Instruct** | | | | | |
| CoT Distillation + RAG | 59.6 | 53.2 | 5.6 | 4.5 | 30.7 |
| Agent Distillation | 60.2 | 65.4 | 15.6 | 7.0 | 37.1 |
| *SCoRe-RL* | 69.2 | 70.2 | 18.0 | 11.0 | 42.1 |

# D. Additional Experiment Results

**More baselines in reasoning benchmarks.** Table 7 compares CoT Distillation, Agent Distillation, and our proposed *SCoRe-RL*, with all methods evaluated using the exact-match metric on math reasoning benchmarks, following (Kang et al., 2025). CoT Distillation trains smaller models to replicate step-by-step reasoning traces generated by large LLMs via Chain-of-Thought prompting. While simple and effective, its reliance on static reasoning limits generalization to novel knowledge or precise computation and can lead to hallucinations. Agent Distillation (Kang et al., 2025) transfers both reasoning and tool-use skills from LLM agents by imitating complete `[Thought-Action-Observation]` trajectories. Its training process is similar to our BC baseline in the main experiments (Tables 1 and 2), but introduces additional enhancements: a first-thought prefix to produce higher-quality trajectories and a self-consistent action generation strategy to improve inference robustness.

Agent Distillation has been shown to outperform vanilla CoT Distillation by incorporating structured, agent-style reasoning into the distillation process. *SCoRe-RL* further exploits correction-based trajectories and RL to deliver substantial performance gains. On `Qwen2.5-7B-Instruct`, *SCoRe-RL* attains an average score of 49.3, a +7.5 improvement over Agent Distillation (41.8), with notable gains on challenging benchmarks such as AIME (+5.6) and OlymMATH (+4.6). On `Qwen2.5-3B-Instruct`, it raises the average score to 42.1, +5.0 over Agent Distillation (37.1). These results demonstrate that *SCoRe-RL* enables small models to achieve significant improvements beyond existing distillation baselines,

*Table 8.* Additional baseline results on deep search. We report stronger baselines, including On-Policy Distillation (OPD), a PRM-style baseline with step-level rewards, and ReLIFT. For OPD, we use Qwen3-32B, the largest dense Qwen3 model, as the teacher for the Qwen3-8B student, since the teacher and student typically must come from the same model family.

| Method | GAIA | HLE | XBench | WebWalker | Avg. |
|---|---|---|---|---|---|
| **Student: Qwen3-8B (Non-thinking)** | | | | | |
| OPD | 30.1 | 9.2 | 20.0 | 37.5 | 24.2 |
| BC + GRPO (Step Reward) | 34.0 | 9.6 | 23.0 | 35.5 | 25.5 |
| ReLIFT | 35.0 | 9.4 | 23.0 | 39.5 | 26.7 |
| *SCoRe-RL* | **40.8** | **11.0** | **27.0** | **43.0** | **30.5** |

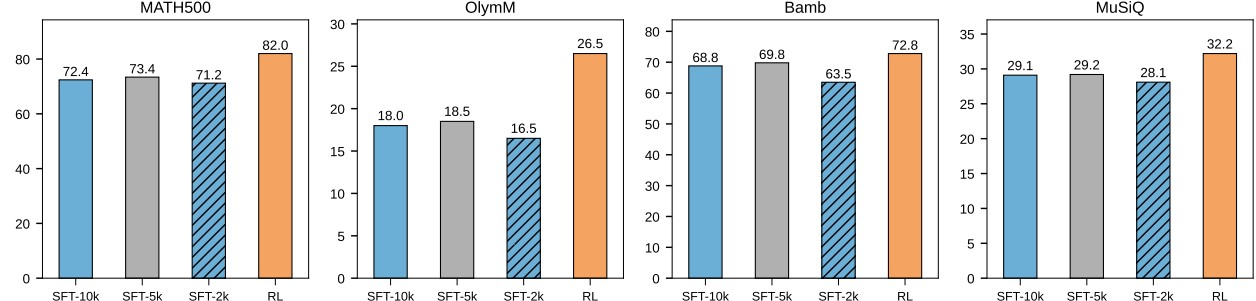

*Figure 4.* Performance of models trained with SFT on MPS-generated data (data scales: 10K, 5K, 2K), compared to an RL-trained model. For math tasks, performance is measured as agreement between generated and reference answers, using `Qwen2.5-72B-Instruct`; QA tasks are evaluated using the F1 score for answer similarity. The evaluation protocol matches that used in the main paper.

narrowing the gap to much larger teacher models.

**More baselines in deep search benchmarks.** We additionally implement three stronger baselines, and compare with them on the deep search benchmarks.

First, we implement a *PRM-style* baseline (**BC + GRPO (Step Reward)**), where the teacher evaluates each Thought–Action–Observation step following the general setup of Zhang et al. (2025). The reward is defined as the proportion of steps judged to move toward the correct solution, providing denser teacher-guided supervision than binary task-success rewards.

Besides, we implement *On-Policy Distillation* (OPD) (Lu & Lab, 2025). Since OPD typically requires the teacher and student to come from the same model family, for the deep search setting with a Qwen3-8B student we use the largest dense Qwen3 model as the teacher. This teacher is slightly stronger than the main teacher used in *SCoRe*.

We also implement *ReLIFT* (Ma et al., 2026), an interleaved RL–SFT baseline originally proposed for reasoning tasks. Since our setting involves agentic reasoning with tool use, the number of hardest samples, i.e., queries on which all rollout attempts fail, is relatively small. To provide ReLIFT with sufficient online SFT data, we use twice the training data, with seed question–answer pairs drawn from ToolStar, the same source as ours.

The results in Table 8 suggest that teacher guidance alone is not sufficient. The PRM-style baseline already provides dense teacher supervision at every step, yet it still underperforms SCoRe-RL. As illustrated in Figure 1, *SCoRe* retains teacher corrections only when they are validated by the student's success, and further combines such verified guidance with short-horizon RL. We believe this coupling of verification and localized policy optimization is a key advantage of *SCoRe*.

On-policy distillation also underperforms SCoRe. A likely reason is that it directly imitates teacher trajectories that may not be well matched to the student's current capabilities. Although this mismatch may be reduced with substantially more data and compute as the student improves, under our training budget *SCoRe* remains more effective.

*SCoRe-RL* also outperforms ReLIFT. We suppose that short-horizon RL provides a more stable optimization signal in our setting, whereas ReLIFT may require a larger training budget and more ReLIFT-style hard samples to achieve comparable performance.

**RL vs. SFT.** In the main experiments, *SCoRe* was first initialized via BC on 5K samples, followed by SFT on 5K MPS-

*Figure 5.* Performance comparison of SCoRe-SFT, SCoRe-RL, and a DPO baseline. While DPO uses the same MPS-generated data as *SCoRe-RL* in a preference-learning formulation, it yields only marginal gains over SCoRe-SFT. The evaluation protocol matches that used in the main paper.

generated samples and RL on another 5K samples. We further examine whether using all data for SFT yields additional gains and assess the impact of SFT data scale. As shown in Figure 4, for both mathematical and search tasks, MPS-generated data improves performance; however, increasing the SFT set from 5K to 10K offers no clear benefit and in some cases slightly degrades results. In contrast, RL training consistently delivers substantial improvements across tasks, indicating that SFT alone is insufficient, while RL is particularly effective for agentic-style tasks.

We also compare our *SCoRe-RL* against a DPO baseline using the same MPS-generated data. As shown in Figure 5, DPO yields only marginal improvements over *SCoRe-SFT*, whereas *SCoRe-RL* achieves large gains on all four benchmarks. This is because DPO requires high data diversity. Specifically, DPO requires multiple alternative completions under the same prefix to fully leverage preference learning. By contrast, *SCoRe*'s short-horizon RL directly optimizes the policy using true task-level returns, avoids reliance on large and diverse preference pairs, and stabilizes training by rolling out from the verified prefix preceding the error.

# E. Prompts used in Mentored Problem-Solving

Figure 6 and 7 illustrate how prompts are used to guide the teacher to find the wrong step and correct it, while Table 9 and 10 present examples of MPS data generation.

---

FIND_WRONG_STEP_PROMPT

You are a precise evaluator. Your task is to analyze a step-by-step reasoning process (step 1 is a first-thought prefix, which is an overall idea for solving this problem, and the remaining steps are the "thought-code cycle") and determine if the final answer is correct.

**INSTRUCTIONS:**
1. Review the entire "Thought-Code Cycle" history provided below.
2. Compare the final answer to the true answer.
3. **If the answer is correct:**
   - The "error_analysis", "correction_start_step" and "correction_suggestion" fields in your JSON output should be null.
4. **If the answer is incorrect:**
   - **Pinpoint the exact step** in the cycle where the error occurred in "correction_start_step".
   - **Explain the nature of the error** (e.g., "The calculation in step 1 was correct, but the rounding in step 2 was incorrect.").
   - **Suggest a specific correction** for the erroneous step.
5. Conclude your response with a single JSON object on a new line. The JSON object must contain the following keys: "is_correct", "error_analysis", "correction_start_step", "correction_suggestion".

**EXAMPLE (Incorrect Answer):**
Question: What is 10 / 3, rounded to the nearest integer?
Correct Answer: 3
Thought-Code Cycle:
Step 1: <first_thought>I will use the math packages of python to solve the problem.</first_thought>
Step 2:
Thought: I will divide 10 by 3 and then round the result up.
Code:
```python
import math
result = math.ceil(10 / 3)
print(result)
```
Observation: 4
Step 3:
Thought: I will provide the final answer.
Code:
```python
final_answer_print("\\boxed{{result}}")
```
Observation: 4

**YOUR RESPONSE:**
```json
{
    "is_correct": false,
    "error_analysis": "...",
    "correction_start_step": 2,
    "correction_suggestion": "..."
}
```

*Figure 6.* Prompt template for the teacher to find the wrong step

CORRECT_PROMPT

**Question:** {original_query}
**Previous Context:** {previous_context}
**Error Step:** {error_step}
**Failed Experience:** {failed_experience}

Based on the above failure analysis, generate the next thought and code to correct the mistake. Provide only one step of thought and code, not the complete solution.
**IMPORTANT:**
1. Always provide a 'Thought:' sequence, and a 'Code: "'python' sequence ending with ''"' sequence, else you will fail. For math problems that are not multiple-choice, always output the final answer using LaTeX \boxed format. Provide the exact value (e.g., \\boxed{{\\frac{{19}}{{14}}}}, \\boxed{{\\sqrt{{2}}}}), not a decimal approximation (e.g., \\boxed{{0.642857}}, \\boxed{{1.41}}).
2. Write simple and short code for each step, and don't try to solve the whole problem in one go. A good code block should only do one thing and include only a brief comment that explains it.
3. You are given a failed experience, please pay attention to it! BUT Don't ACT LIKE YOU'RE CORRECTING A MISTAKE.
4. In the end you have to return a final answer, use the `final_answer_print` tool to print it, or you will be in an endless loop!
Please only give the 'Thought and Code' for the current cycle.

*Figure 7.* Prompt template for the teacher to correct the wrong step.

*Table 9.* An example of **MPS** data generation for a math problem, with tagged thoughts (`<thought>`), Python code, observations, and final answer (`<answer>`). Teacher corrections are highlighted in orange.

---

**Question:** In the figure, $BA = AD = DC$ and point $D$ lies on segment $BC$. The measure of $\angle ACD$ is 22.5°. What is the measure of $\angle ABC$?

**Labeled Answer:** 45°

---

### Step 0 (Student overall idea)

`<first_thought>` To solve this problem, I need to analyze the given geometric configuration and use the properties of isosceles triangles and angle relationships. Since $BA = AD = DC$, triangles $ABD$ and $ADC$ are isosceles. I can use $\angle ACD = 22.5°$ to find the other angles in $\triangle ADC$, and then use properties of isosceles triangles to find $\angle ABC$. `</first_thought>`

### Step 1 (Student incorrect attempt)

`<thought>` Since $BA = AD = DC$, triangles $BAC$, $ABD$, and $ADC$ are isosceles. I can use the fact that $\angle ACD = 22.5°$ to find the other angles in $\triangle ADC$. `</thought>`

```
import math
# Given angle ACD = 22.5 degrees
angle_ACD = 22.5
# Since AD = DC, triangle ADC is isosceles with angle ADC = angle ACD
angle_ADC = angle_ACD
# The sum of angles in a triangle is 180 degrees
angle_CAD = 180 - 2 * angle_ACD
print(angle_CAD)
```

Observation: 135.0

### Teacher correction at Step 1:

`<thought>` Given $\angle ACD = 22.5°$ and $AD = DC$, it follows that $\angle CAD = \angle ACD = 22.5°$. `</thought>`

```
# Given angle ACD = 22.5 degrees
angle_ACD = 22.5
# Isosceles triangle ADC: AD = DC, so angle DAC = 22.5 degrees
angle_CAD = angle_ACD
print(angle_CAD)
```

Observation: 22.5

---

### Step 2 (Student continues after correction)

`<thought>` Since $BA = AD$, $\triangle BAD$ is isosceles with $\angle ABD = \angle ADB$. Given $\angle ADC$, we find $\angle ADB$, then determine $\angle ABC = \angle ADB$. `</thought>`

```
# Find angle ADC
angle_ADC = 180 - angle_CAD - angle_ACD
# Angles ADC and ADB are supplementary
angle_ADB = 180 - angle_ADC
# Triangle BAD is isosceles (BA = AD): angle ABC = angle ADB
angle_ABC = angle_ADB
final_answer_print(angle_ABC)
```

Observation: 45.0

---

`<answer>` 45.0 `</answer>`

---

*Table 10.* An example of **MPS** data generation for a music trivia question, with wrong step and teacher correction. Teacher corrections are highlighted in orange.

---

**Question:** What is the award that the performer of the song "Smoke On The Water" earned?
**Labeled Answer:** Rock and Roll Hall of Fame

---

**Step 0 (Student overall idea)**
`<first_thought>` To find out the award that the performer of the song "Smoke On The Water" earned, I will first need to identify the performer of the song. Once I have the name of the performer, I can then search for any awards they have won. I will use a search tool to gather this information. `</first_thought>`

---

**Step 1 (Correct prefix)**
`<thought>` I need to find out who performed the song "Smoke On The Water" and then search for any awards they have won. `</thought>`

```
web_search("performer of Smoke On The Water")
```

Observation: Web Pages: ... (The search results show that "Smoke On The Water" was performed by Deep Purple, with Ian Gillan as lead singer).

---

**Step 2 (Correct prefix)**
`<thought>` The search results indicate the performer is Deep Purple (lead singer Ian Gillan). Next, I need to search for any awards that Ian Gillan or Deep Purple have won. `</thought>`

```
web_search("awards won by Ian Gillan")
```

Observation: Web Pages: ... (Found references to Ivor Novello Award for International Achievement, Legend Award at 2008 World Music Awards, and Classic Rock Innovator Award).

---

**Step 3 (Student incorrect attempt)**
`<thought>` The most relevant award is the Ivor Novello Award for International Achievement, received by Ian Gillan, Roger Glover, and Ian Paice of Deep Purple. I will use this as the final answer. `</thought>`

```
final_answer_print("Ivor Novello Award for International Achievement")
```

Observation: Ivor Novello Award for International Achievement(Incorrect in context)

---

**Teacher correction at Step 3:**
Error Analysis: While Deep Purple members did win the Ivor Novello Award, the question asks for the award that the performer of "Smoke On The Water" earned. However, the band has not received this award in any context directly related to "Smoke on the Water".
`<thought>` The most significant award for Deep Purple, especially tied to the legacy of "Smoke On The Water", is their Rock and Roll Hall of Fame induction in 2016. I will verify this. `</thought>`

```
web_search("Deep Purple Rock and Roll Hall of Fame")
```

Observation: Web Pages: ... (Confirmed — Deep Purple inducted into the Rock and Roll Hall of Fame in 2016).

---

**Step 4 (Student continues after correction)**
`<thought>` Based on confirmation, the correct award is Rock and Roll Hall of Fame. `</thought>`

```
final_answer_print("Rock and Roll Hall of Fame")
```

Observation: Rock and Roll Hall of Fame

---

`<answer>` Rock and Roll Hall of Fame `</answer>`

