# OpenReview forum: "Student-Centered Distillation Narrows the Agentic Gap Between Small and Large LLMs"
_ICML.cc/2026/Conference — ICML 2026 regular_

### Official Review · Reviewer_J2SN · 2026-03-02

**Soundness:** 3
**Presentation:** 3
**Significance:** 3
**Originality:** 2
**Overall Recommendation:** 3
**Confidence:** 4

**Summary:**

The paper introduces SCoRe (Student-Centered one-step Reinforcement), a distillation framework designed to bridge the agentic performance gap between small and large Large Language Models (LLMs). To address the compounding errors typical of pure Behavior Cloning (BC), SCoRe employs a Mentored Problem-Solving (MPS) phase where the student generates a trajectory, and the teacher intervenes only to correct the first error. The student resumes from this correction, producing data aligned with its own capabilities. This data is used for a correction-based Supervised Fine-Tuning (SFT) phase, followed by a short-horizon Reinforcement Learning (RL) phase that utilizes key-step rewards to encourage the student to reproduce the teacher's fix or avoid the original mistake. The authors demonstrate that SCoRe enables smaller models (e.g., 7B) to achieve performance comparable to teacher models (e.g., 72B) across various benchmarks.

**Compliance With Llm Reviewing Policy:**

Affirmed.

**Final Justification:**

My final recommendation remains **Weak Reject**. While the rebuttal partially addressed my concerns, it did not fully resolve the main issues around originality and soundness. In particular, I now better understand that the claimed novelty lies in using first-error localization to define the RL rollout start and localized reward, rather than in the correction-based data construction itself. This is a meaningful distinction, but I still find the overall method somewhat incremental relative to prior correction/regeneration-style approaches. I remain concerned that the theoretical claims are somewhat idealized, and that some of the stronger interpretations of the hard-case results are not yet fully supported.

**Key Questions For Authors:**

Q1. Inconsistent Backbone Selection Across Benchmark Categories:
In Table 1, the primary student models evaluated are Qwen2.5-7B-Instruct and Qwen2.5-3B-Instruct. However, in Table 2 for the Deep Search benchmarks, the student model suddenly shifts to Qwen3-8B. The caption in Table 1 explains that Qwen3-8B was excluded from reasoning tasks due to potential data leakage risks (October 2024 data cutoff). While this justifies excluding Qwen3-8B from Table 1, it absolutely does not justify omitting Qwen2.5-7B-Instruct from Table 2. Changing the base model architecture across different benchmark suites breaks the empirical consistency of the paper. It makes it impossible for the reader to determine whether the strong performance on Deep Search is primarily driven by the SCoRe framework or simply by the inherently superior capabilities of the newer Qwen3-8B model. Could you please provide the performance results of Qwen2.5-7B-Instruct or Qwen2.5-3B-Instruct on the Deep Search benchmarks (GAIA, HLE, XBench, WebWalker) to ensure an apples-to-apples evaluation across all tasks? If the SCoRe framework is robust, it should exhibit consistent performance gains on the same Qwen2.5 backbone across both reasoning and search domains.

**Limitations:**

There is no limitation section, which should be added.

**Strengths And Weaknesses:**

### Strengths
* **Well-Motivated Problem:** The paper accurately identifies and tackles the compounding error problem inherent in standard imitation learning/behavior cloning for long-horizon LLM agents.

* **Intuitive Data Pipeline:** The "student explores, teacher corrects" paradigm is an elegant way to generate capability-matched training data. Localizing deficiencies effectively bridges the gap between the student's current capacity and the teacher's optimal policy.

* **Clear Empirical Results:** The proposed method demonstrates substantial improvements over standard BC and trajectory-level RL (GRPO/ARPO) across multiple benchmarks.

### Weaknesses

**W1. Insufficient Comparison with Closely Related Work**
The paper claims novelty in using a large teacher to correct a smaller student's trajectory and using teacher-guided rewards. However, it lacks discussion and comparison with highly relevant existing literature. For instance, the core idea of identifying the first error in a reasoning chain and performing a correction-based restart/regeneration is heavily explored in works like LEMMA [1] (also identified the first erroneous step in the reasoning chain, and then a corrective restart/regeneration is performed from that point) and [2]. Furthermore, using a large teacher model to provide step-level rewards for a student during RL is conceptually identical to Process Reward Models (PRMs), such as R-PRM [3] or [4]. The authors must clearly contextualize their work and distinguish their contributions from these established lines of research.

**W2. Lack of Comprehensive Baselines for the Two-Stage Pipeline**
SCoRe employs a two-stage pipeline consisting of an SFT phase followed by an RL phase. However, the chosen baselines fail to effectively isolate the source of the performance gains. To prove the superiority of the proposed SCoRe data construction and RL mechanisms, the authors should include a "Behavior Cloning + GRPO" baseline. Comparing SCoRe-RL against pure BC or pure trajectory-level RL (without BC initialization) is confounding, as SCoRe benefits from both SFT initialization and targeted RL exploration. Additionally, the paper should compare its "key-step reward"  against a standard PRM baseline (which provides dense, step-by-step process rewards) to validate why providing key-step rewards is more effective.

**W3. The "Hard-to-Teach" Paradox Lacks Qualitative Evidence**
The paper claims that retaining "Hard-to-Teach" data (instances where the task remains unsolved even after the 72B teacher's multiple one-step corrections ) allows the 7B student to surpass the teacher's strategy space via RL exploration. This claim is highly counter-intuitive. If a powerful 72B teacher cannot guide the student to a solution through targeted corrections, it is unconvincing that a 7B model would suddenly achieve an "epiphany" and find the correct path relying solely on sparse environmental rewards (final task success). Without deep qualitative analysis or case studies, it is impossible to tell whether the student actually learned superior strategies, or if the performance gain is merely an artifact of testing variance, reward hacking, or the student simply bypassing specific prompt constraints (formatting rules) that had bottlenecked the teacher.

**W4. Credit Assignment Noise in the Format Reward Design**
The rationale behind the format reward design is questionable. The paper defines the format reward $\eta = R_{format}$ under the mutually exclusive condition that the student's action is different from both its original error and the teacher's correct action ($a_k \ne a_k^{orig} \land a_k \ne a_k^{\pi_E}$). While models like DeepSeek-R1 grant format rewards strictly for adhering to correct formatting templates (e.g., generating proper <think> tags), SCoRe's condition essentially rewards the model for simply "making a different mistake." In long-horizon reasoning tasks, this introduces severe credit assignment noise. The model is actively incentivized to output arbitrary or illogical actions just to satisfy the inequality, rather than being guided toward a logically sound path.

**W5. Flaws in Theoretical Justifications**
The theoretical justifications in Section 3 rely on overly strong assumptions and ignore critical RL fundamentals:

- Unrealistic Assumptions in Theorem 3.2: The authors assume that the student's per-step error rate is bounded by $\epsilon$ under its own induced state distribution $d_t^{\hat{\pi}}$ ($\mathbb{P}_{s \sim d_t^{\hat{\pi}}}[\hat{\pi}(s) \ne \pi_E(s)] \le \epsilon$). This trivially assumes away the core challenge of imitation learning. The very reason BC suffers from compounding errors is Covariate Shift: while the error rate ($\epsilon$) might be small on the expert's distribution, it can become unbounded once the student deviates into its own distribution.

- Missing Bias-Variance Tradeoff in Theorem 3.3: The authors prove that shortening the rollout horizon (from $H$ to $H-k+1$) reduces variance. Mathematically, it is trivial that a sum of fewer bounded random variables has a smaller variance upper bound than a sum of more variables. More critically, the authors completely ignore the Bias-Variance Tradeoff. Truncating trajectories at a verified prefix and injecting a heuristic key-step reward inevitably introduces strong bias into the policy gradient. A theoretical analysis that exclusively highlights variance reduction while remaining entirely silent on the introduced bias is incomplete and mathematically misleading.

**W6. Unaccounted Computational Overhead in Data Generation**
The authors frame their work as a cost-reduction strategy for deployment. However, the Mentored Problem-Solving (MPS) phase is computationally expensive. It requires the student to generate a full trajectory, the 72B teacher to review it, pinpoint the error, generate a correction, and then the student to roll back and resume generation. This process can loop multiple times per query. Comparing SCoRe to BC using only the final "trajectory count" (data budget)  is an unfair comparison, as SCoRe consumes exponentially more FLOPs or API calls during the data construction phase.

[1] LEMMA: Learning from Errors for MatheMatical Advancement in LLMs

[2] Learning to Check: Enhancing Self-Correction Capabilities in Large Language Models for Reasoning Tasks

[3] R-PRM: Reasoning-Driven Process Reward Modeling

[4] The Lessons of Developing Process Reward Models in Mathematical Reasoning

---

> ### Author Rebuttal · Authors · 2026-03-29
>
> Thanks for your careful review and valuable suggestions!
> # W1
> - Our theorems do not hold in LEMMA. SCoRe is strictly student-centered:  the teacher fixes only the first error and the student generates the remaining, so in Eq. 6 of the proof we use $s \sim d_t^{\hat{\pi}}$, meaning states are sampled from student distribution. In LEMMA, teacher instead of student generates the remaining trajectory, so states are not from the student distribution, **thus the $O(H^2)$ to $O(H)$ derivation (Eq.16–23) does not hold**.
> - SCoRe uses verifiable step rewards via our unique MPS ( teacher guidance is kept only if student succeeds), PRM lacks such important verification.
> # W2
> - Sorry for the confusion. All RL baselines in Table 1/2 already include SFT initialization; they are Behavior Cloning + GRPO/ARPO and already follow a two-stage pipeline.
> - We add a PRM-style baseline where the teacher evaluates each Thought-Action-Observation step, following ideas and prompts from `The Lessons of Developing Process Reward Models in Mathematical Reasoning`. The teacher judges if each step is valuable, and final reward is a proportion of these steps, providing denser teacher guidance.
>
> Model:Qwen3-8B
> ||GAIA|HLE|XBench|WebWalker|Avg|
> |-|-|-|-|-|-|
> |Step Utility Reward|34.0|9.6|23.0|35.5|25.5|
> |SCoRe-RL|40.8|11.0|27.0|43.0|30.5|
>
> It performs worse than SCoRe‑RL, since PRMs rely on subjective teacher judgments, while SCoRe uses verified step rewards (Fig. 1 suggests the teacher guidance is verified by the student's success) together with short‑horizon RL.
> # W3
> Yes, your suggestion is insightful! And **we already have these experiments in Fig. 3**.
> - The 7B model does not suddenly discover the correct path; its policy evolves gradually:  SFT on MPS data already shifts the student’s capability, and the subsequent RL phase is on-policy, so the policy continues to change.
> - This is reflected in Fig. 3(b). After SFT on MPS data, performance on hard subsets already improves (Math: 0 to 17.3%, Search: 0 to 20.4%). RL further improves it (24.3%, 26.3%).
> - Qualitatively, early failures often occur when the student lacks stable tool use or long-horizon reasoning. As SFT and on-policy RL improve these gradually, some cases become solvable.
> We will revise the wording to clarify that RL enables exploration beyond the teacher's strategy space, rather than implying the student surpasses the teacher.
> # W4
> Sorry for the confusion. Our format reward is essentially the same as DeepSeek‑R1’s, but slightly stricter:
> - DeepSeek‑R1: format-correct
> - Ours: format-correct and not repeating the error (Checking whether a step repeats the error requires output to be format-correct).
>
> Since our reward is a stricter version of DeepSeek‑R1’s, and their format reward has little impact on optimization, our design typically cannot be worse because it's stricter.
> # W5
> - Your observation is insightful, but **covariate shift typically arises in imitation learning: trained on expert states but deployed on student states. In contrast, SCoRe is not an imitation learning framework, and is trained directly on student states with corrections, which mitigates it**.
>
> And **we do not assume the bound $ε \in [0,1]$ is small**: it is not removed from the analysis and appears throughout the proof. Thm 3.2 is a conditional result showing an improvement from $O(H^2ε)$  to $O(Hε)$, $ε$ is never overlooked.
>
> Our proof follows the classic DAGGER: `A Reduction of Imitation Learning and Structured Prediction to No-Regret Online Learning(2011)`, whose conclusions have been widely adopted.
> - For Thm 3.3, thank you for raising it! Like any traditional theorem, variance reduction must introduce bias, and we focus on variance reduction. In our setting bias can be bounded by $O(k + \delta(H-k+1))$ in the undiscounted case $γ=1$, where $\delta$ denotes reward-shaping bias from the key-step reward.
> We will clarify these points in the paper and are happy to provide further details during the discussion.
> # W6
> Thanks for the valuable suggestion!
> - **Luckily, we already include compute‑matched comparisons in our ablation study**. In Table 3, the rows RL w/o short-horizon and RL w/o key-step rewards both use MPS-generated data and the same GRPO pipeline, meaning they have comparable compute with ours.
> - In the response of W2, we also add a stronger PRM baseline, which needs dense compute. Please also see our response to Reviewer 41Zm (W2) for additional analysis.
> # Question
> Thanks for the suggestion!
> Deep Search benchmarks are harder. When evaluating Qwen2.5‑7B on these tasks,  success rates are low for all methods:
>
> Model:Qwen2.5-7B-Instruct
> ||GAIA|HLE|XBench|WebWalker|Avg|
> |-|-|-|-|-|-|
> |GRPO|22.3|6.2|19.0|20.0|16.9|
> |SCoRe-RL|26.2|7.4|23.0|25.5|20.5|
>
> Although SCoRe still improves over GRPO, the low-success regime makes it hard to distinguish capability gains from random search hits. Therefore we report main results with the stronger Qwen3‑8B. We will add the 7B results to the appendix.

---

> > ### Author Rebuttal · Reviewer_J2SN · 2026-04-05
> >
> > Thank you for the detailed rebuttal and additional experiments. However, after carefully reviewing the response, I believe several of my core concerns remain insufficiently addressed.
> >
> > **W1**: The rebuttal mainly emphasizes theoretical differences to distinguish SCoRe from prior work, such as LEMMA and related correction-based frameworks. However, my concern is primarily about algorithmic novelty, not only theoretical analysis. Specifically, prior works such as LEMMA and *Learning to Check* already explore:
> > - identifying the first erroneous step in a reasoning trajectory,
> > - restarting or regenerating from that point,
> > - and leveraging correction-based supervision.
> >
> > From a pipeline perspective, SCoRe appears closely related to this family of methods, with the main extension being the addition of an RL phase. The rebuttal does not clearly articulate what fundamentally differentiates SCoRe’s data construction or training pipeline from these approaches.
> >
> > Similarly, the distinction from Process Reward Models (PRMs) is described as “verification vs. subjective judgment.” However, recent PRM-style works (e.g., R-PRM, and discussions in *The Lessons of Developing Process Reward Models in Mathematical Reasoning*) can also incorporate verifiable signals (e.g., execution-based correctness). It remains unclear whether the improvement comes from:
> > - reward sparsity vs. density,
> > - key-step localization,
> > - or the verification mechanism itself.
> >
> > A clearer positioning relative to these lines of work is still needed.
> >
> > **W3**: The rebuttal clarifies that improvements on hard cases arise gradually through SFT + on-policy RL, which is helpful. However, the core issue remains unresolved: If a 72B teacher fails to guide the student to a solution even with multiple targeted corrections, it is unclear why a 7B model, trained with sparse rewards, would succeed.
> >
> > The current response provides aggregate performance improvements (Fig. 3), but lacks qualitative evidence. In particular, it would be important to show:
> > - concrete examples where the student succeeds while the teacher fails,
> > - and what kind of reasoning patterns or strategies are discovered beyond the teacher’s behavior.
> >
> > Without such analysis, it is difficult to rule out alternative explanations such as reward exploitation, stochastic success, or differences in evaluation conditions.
> >
> > **W5**: I appreciate that the rebuttal acknowledges the bias introduced by short-horizon rollouts and provides an informal bound. However, the main concerns remain:
> >
> > - **Theorem 3.2:** The key issue is not whether $\epsilon$ appears in the bound, but whether it remains bounded under the student-induced distribution $d_t^{\hat{\pi}}$. This is precisely where covariate shift becomes problematic in imitation learning (Ross et al., 2011). The rebuttal does not clarify under what conditions this assumption holds.
> >
> > - **Theorem 3.3:** While variance reduction is shown, the bias–variance tradeoff is not rigorously analyzed. The provided bound is not formally derived in the paper, nor empirically validated.
> >
> > Overall, the theoretical section still appears incomplete and somewhat idealized relative to the actual training dynamics.
> >
> > **W6**: The rebuttal discusses compute-matched comparisons for RL variants, but this does not address the original concern regarding **data construction cost**.
> >
> > MPS phase requires:
> > - full student trajectory generation,
> > - teacher review over the entire trajectory,
> > - potentially multiple correction cycles with rollback and regeneration.
> >
> > This process is significantly more expensive than standard behavior cloning data generation. The paper’s comparison based on “trajectory count” does not reflect the true computational cost. Quantitative reporting (e.g., number of teacher calls or total FLOPs per sample) would be necessary for a fair comparison.

---

> > > ### Author Response · Authors · 2026-04-07
> > >
> > > Thanks for your thoughtful and constructive feedback!
> > > # W1
> > > In data construction, we agree SCoRe shares the high-level "identify an error and continue" structure as LEMMA-style methods(A difference is that in SCoRe the student, rather than teacher resumes, which is basis of our theoretical analysis).
> > >
> > > At algorithmic level, the main difference is not the data format itself, but how first-error correction is turned into training:
> > > - prior correction-based methods use the corrected trajectory as the training unit, while SCoRe-RL uses (verified prefix, short rollout, localized reward);
> > > - Prior methods optimize next-token imitation/regeneration, while SCoRe optimizes policy improvement at the first failure state.
> > >
> > > Therefore, SCoRe is not simply a correction pipeline plus a generic RL phase: it uses the first error step to decide where rollout starts and where reward is given, i.e., RL state setup, shorter rollout, and local credit assignment.
> > >
> > > Regarding PRMs, we agree that if PRM-style methods also use verifiable signals, then verification itself is not the novelty. The key difference is how verification is used: PRM-style methods aim to learn a process scorer, while SCoRe does not train such a scorer. Instead, SCoRe uses verification to find the first error step on student-generated trajectories, and then sets rollout and reward around that point. So the gain does not come from verification alone, but from verification-based first-error localization, which decides both rollout shortening and local rewards.
> > > # W3
> > > Thank you for the suggestion. We agree previous rebuttal mainly gave quantitative evidence(Fig. 3), while this question also needs qualitative analysis in some cases.
> > > - One example is: "What is the best award that the performer of Smoke On The Water earned?"
> > >
> > > The teacher's repeated corrections roughly follow: Identify the performer as Deep Purple/Ian Gillan; Search for multiple awards won by the performer; After retrieving Ivor Novello Award for International Achievement, output it.
> > >
> > > The issue is not missing knowledge, but coarse correction: there are many retrieved awards and the student can't tell which one best matches question. In practice, the student often fails here and stays stuck in "output the first retrieved award."
> > >
> > > After RL, successful student trajectories often add a verification step. Instead of outputting the first retrieved award, the student issues an extra query such as "Deep Purple Rock and Roll Hall of Fame", checks which award best matches the question, and then outputs Rock and Roll Hall of Fame. So the gain is not just imitating teacher corrections, but learning to add validation between "finding a candidate" and "giving the answer".
> > >
> > > - We also manually analyzed 50 previous hard but now solvable cases, and found why the RL-trained student succeeded:
> > >
> > > rewriting retrieval strategy: 20/50, reordering subproblem decomposition: 14/50, adding a verification step: 10/50, other(format): 6/50.
> > >
> > > This suggests the gains on hard cases mainly come from interpretable strategy changes, especially finer-grained decomposition and verification rather than simply following teacher edits.
> > >
> > > Regarding alternative explanations: The teacher and student are evaluated under the same tool environment and protocol, so this is not due to different evaluation settings. This is also inconsistent with reward exploitation, since successful trajectories usually involve more retrieval/verification steps rather than shortcut behaviors.
> > >
> > > The patterns are consistent across manually inspected cases, making pure chance less plausible.
> > >
> > > Such qualitative case studies and existing experiments show how RL helps the student find strategies beyond the teacher's corrections.
> > > # W5
> > > We appreciate this observation and agree that the assumption should be stated more carefully. In the revision, we will clarify that Thm. 3.2 is conditional, in line with DAgger’s assumption-style guarantee: if the train-test mismatch in the student distribution $d_t^{\hat{\pi}}$ is bounded, the proof holds. This assumption is too strong for pure imitation learning, where training uses $d_t^{\pi_E}$. However, SCoRe trains on $d_t^{\hat{\pi}}$ with minimal teacher corrections, aligning supervision with $d_t^{\hat{\pi}}$ at inference. Under this setting, we believe the result has limitations but still provides useful insight.
> > >
> > > For the bias bound in Thm. 3.3, we provide a detailed proof in Figures: https://anonymous.4open.science/r/Proof-75DF.
> > > # W6
> > > Thanks for raising this point!  We measured the average data construction token usage per teacher intervention (prompt + output):
> > > - Math tasks: 811 tokens with one intervention; 1275 with two.
> > > - Search tasks: 1260 tokens with one intervention; 2046 with two.
> > >
> > > In practice, most trajectories require only one teacher intervention(as Fig. 3a shows), so the number of teacher calls per sample is close to one. Since the teacher corrects only the first error rather than every step, the supervision cost remains moderate.

---

### Official Review · Reviewer_NtqZ · 2026-03-09

**Soundness:** 2
**Presentation:** 3
**Significance:** 3
**Originality:** 3
**Overall Recommendation:** 4
**Confidence:** 3

**Summary:**

This paper introduces SCoRe, a student-centered training framework for improving student models on agentic tasks through teacher guidance. Unlike conventional distillation methods that rely primarily on teacher-generated trajectories, SCoRe lets the student first produce trajectories, which are then reviewed and corrected by the teacher. Building on this principle, the paper proposes two training stages: SCoRe-SFT, which fine-tunes the student on teacher-corrected trajectories, and SCoRe-RL, which further improves the student via short-horizon reinforcement learning. Experiments on 12 challenging benchmarks show that SCoRe substantially improves student performance and significantly narrows the gap to a 72B teacher model.

**Compliance With Llm Reviewing Policy:**

Affirmed.

**Final Justification:**

The authors' rebuttal has addressed my concerns regarding to the paper. Thus I change the score to weakly accept.

**Key Questions For Authors:**

1. What is the performance of SCoRe-RL without warm start (SCoRe-SFT)? Would it still outperform the RL baselines?
2. What is the performance of on-policy distillation on these tasks?
3. The training data of SCoRe-RL is generated in an off-line fashion, right? What's the scalability of this process?

**Limitations:**

Limitations of the work are not discussed in the paper.

**Strengths And Weaknesses:**

Strength:
- The paper is clearly written and well-structured
- The method is well motivated and the idea of student-centered training is interesting
- The use of short-horizon RL is also technically well motivated and represents a meaningful design choice for improving training stability and efficiency.
- The authors have conducted extensive experiments on three categories of datasets
- Experiments demonstrate the effectiveness of the proposed method compared to the baselines.

Weakness:
- **Over-complicated design choice**. The proposed method contains both SFT and RL steps, each with several design choices, making it unclear about which part is crucial for the performance gain. The current ablations are not sufficient to fully disentangle the contribution of each part.
- **Unfair comparison**. The comparison to RL baselines is not entirely apples-to-apples. The SCoRe-RL contains 1) SFT for warm start, and 2) teacher guidance for key-step rewards. However, the RL baselines are run over full trajectories with sparse rewards. This makes the comparison unfair and weakens the effectiveness of SCoRe.  For example, on Mathematical Reasoning and Factual Reasoning problems (table 1), SCoRe-RL + SFT obtains 50.8% average performance, but the RL baseline (APRO) alone achieves 49.3% average performance without any teacher guidance.
- **Missing baselines**. SCoRe contains both SFT and RL steps with the teacher guidance. However, the experiment only contains 1 SFT baseline (Behavior Cloning) and 2 RL baselines (GRPO and ARPO), which are insufficient to demonstrate the effectiveness of SCoRe. Introducing other baselines (such as on-policy distillation) would be very helpful.
- **Insufficient discussion of short-horizon RL**. Beyond reporting final benchmark numbers, the paper does not adequately discuss the trade-offs in training efficiency, annotation cost, and scalability.

---

> ### Author Rebuttal · Authors · 2026-03-29
>
> # W1
> We apologize for the confusion.
> As noted, our method consists of two phases: SFT and RL. The current ablation study in Table 3 already covers most design choices in both phases, but the presentation may be less clear. We reorganize the ablations below to more clearly separate the contributions.
> |Phase|Design|Avg|AIME24|AIME25|MATH500|OlymMath|HQA|2Wiki|MuSiQ|Bamb|
> |-|-|-|-|-|-|-|-|-|-|-|
> |SFT|Initial Distillation (BC init)|41.7|20.0|23.3|70.8|...| More results are reported in Table 3 of the original paper. |
> |SFT|SFT on MPS data (SCoRe-SFT)|45.8|26.7|16.7|73.4|...|
> |RL|RL w/o short-horizon|48.4|30.0|20.0|78.0|...|
> |RL|RL w/o key-step rewards|49.7|33.3|20.0|80.6|...|
> |RL|Full design (SCoRe-RL)|50.8|36.7|26.7|82.0|...|
>
> This organization highlights that the gains come from two sources: capability-aligned SFT data from MPS, and the two RL design components (short-horizon rollouts and key-step rewards). **If there are additional specific ablations the reviewer would like to see, we would be happy to include them**.
> # W2
> Thank you for the insightful comment!
> - First, the **RL baselines in Table 1/2 (GRPO and ARPO) also use SFT warm‑start**. In practice they are trained as Behavior Cloning + RL (GRPO/ARPO), meaning they already include some teacher guidance. However, we acknowledge SCoRe introduces additional teacher guidance through key‑step rewards.
> - Following the reviewer's suggestion, we implemented **two more baselines with more teacher guidance**:
>
> [1] A PRM-style baseline where the teacher evaluates each Thought-Action-Observation step, following ideas and prompts from `The Lessons of Developing Process Reward Models in Mathematical Reasoning`. The teacher judges if each step moves toward the correct solution given the reference solution, and the final reward is a proportion of these steps. This produces reward guided by the teacher, rather than a binary success signal.
>
> [2] On-policy distillation. OPD typically requires the teacher and student to be from the same model family. Since we use dense Qwen3-8B on deep-search benchmarks, we use the largest dense Qwen3 model as the teacher, which is slightly stronger than our main teacher.
>
> Qwen3‑8B as the base model
> |Method|GAIA|HLE|XBench|WebWalker|Avg|
> |-|-|-|-|-|-|
> |Step Utility Reward|34.0|9.6|23.0|35.5|25.5|
> |On-policy Distillation|30.1|9.2|20.0|37.5|24.2|
> |SCoRe-RL|40.8|11.0|27.0|43.0|30.5|
>
> - The results show that teacher guidance alone is insufficient. The PRM-style baseline already provides dense teacher supervision for every step, yet it still underperforms SCoRe-RL. As illustrated in Fig. 1, SCoRe keeps teacher corrections only after they are validated by the student's successful completion, and combines this verified guidance with short-horizon RL, which we believe is the key advantage of our framework.
> # W3
> The on-policy distillation has been added in response to W2.
>
> On‑policy distillation imitates teacher's trajectories that may be misaligned with the student's capabilities. While this gap may diminish with substantially more data and compute as the student's ability improves, under our training budget it underperforms SCoRe.
> # W4
> For annotation cost, we report the average token consumption (prompt + output) of teacher interventions. Using Qwen2.5‑72B‑Instruct as the teacher (locally deployed in our experiments):
> - Math tasks: 811 tokens per sample with one intervention; 1275 tokens with two interventions.
> - Search tasks: 1260 tokens per sample with one intervention; 2046 tokens with two interventions.
> This shows that the teacher supervision required by SCoRe remains moderate in practice.
>
> We also include a training data scale study for RL:
>
> Qwen3‑8B as the base model
> |RL Data Scale|GAIA|HLE|XBench|WebWalker|Avg|
> |-|-|-|-|-|-|
> |2K|35.0|9.2|22.0|39.5|26.4|
> |5K|38.8|8.8|29.0|41.0|29.4|
> |10K|40.8|11.0|27.0|43.0|30.5|
> # Question
> Thank you for the insightful suggestion! Q2 and Q3 have been answered in the response above. For SCoRe-RL without warm start (SCoRe-SFT)
>
> - For the Qwen2.5 series, warm‑start SFT is necessary because the base model has limited tool‑use ability. Without SFT initialization, the model often fails to generate correctly formatted tool calls during RL rollouts. For this reason, all RL baselines in Tables 1/2 (GRPO and ARPO) also include the same warm‑start SFT phase.
> - In contrast, Qwen3 models have stronger native tool‑use ability and can directly enter RL without a warm start. We therefore evaluate SCoRe‑RL without a warm start on the DeepSearch benchmarks:
>
> |Method|GAIA|HLE|XBench|WebWalker|Avg|
> |-|-|-|-|-|-|
> |SCoRe‑RL (use MPS data but w/o SFT)|38.8|10.2|25.0|40.5|28.6|
>
> The results show that SCoRe‑RL can still improve performance without a warm start on Qwen3, while warm‑start SFT provides additional gains by stabilizing early training.

---

> > ### Author Rebuttal · Reviewer_NtqZ · 2026-04-04
> >
> > Thank you for the detailed response and my concerns have been addressed. I've updated my score accordingly.

---

> > > ### Author Response · Authors · 2026-04-05
> > >
> > > We sincerely appreciate your thoughtful and constructive feedback, which has greatly helped improve our work. Thanks for your patience and careful review again!

---

### Official Review · Reviewer_5HeL · 2026-03-12

**Soundness:** 3
**Presentation:** 4
**Significance:** 3
**Originality:** 3
**Overall Recommendation:** 4
**Confidence:** 3

**Summary:**

This work proposes a mixed distillation framework that uses both supervised fine-tuning (SFT) and reinforcement learning (RL) for skill transfer, allowing precise error identification and targeted reinforcement. In particular, the teacher model monitors the student model’s reasoning process and corrects wrong steps as they appear. The student model completes the reasoning process with the teacher model’s assistance. The obtained reasoning responses are then used for SFT. Additionally, RL is performed starting from the step right before the error, shortening the reasoning process and allowing the model to focus on the weaknesses. It is theoretically proved that such training could effectively reduce the variance of the policy. Experimental results showed that the mixed distillation approach is more effective than SFT or RL alone.

**Compliance With Llm Reviewing Policy:**

Affirmed.

**Final Justification:**

This work proposes an approach for agentic distillation and the rebuttal clarifies its differences with related work as well as better performance.

On the other hand, I agree with other reviewers that several components of this work have been proposed before, such as mixing SFT and RL, identifying the first erroneous step, restarting or regenerating from that point, and leveraging correction-based supervision. As such, the main novelty is a creative combination of such components and applying this training pipeline to agentic tasks, which is relatively small.

Considering both the strengths and weaknesses, I keep my current rating.

**Key Questions For Authors:**

I have no particular questions.

**Limitations:**

I suggest discuss the following potential limitations:
- The student model was never trained for this type of interaction and tool use. So, the effectiveness of this approach for other better pretrained models remained unclear.
- This approach requires the teacher model to be good at both generation and verification. Being weak in any of them may result in compromised distillation.
- Distribution mismatch between the teacher and student model may reduce the effect of SFT.

**Strengths And Weaknesses:**

Strengths:
- This work proposes an effective approach that combines the advantages of SFT and RL.
- Comparing to classic SFT and RL, this approach performs more targeted reinforcement on the weaknesses of the models, which may result in more efficient distillation process.
- This approach reduces the variance of models.
- Effectiveness of the approach was well-evaluated. Results clearly showed that it was more effective than SFT and RL alone.

Weaknesses:
- The experiment did not compare this approach with other more sophisticated distillation approaches, such as off-policy RL (e.g., LUFFY), and SFT-RL hybrid approaches (e.g., SRFT).
- Limitations of this work were not discussed.

---

> ### Author Rebuttal · Authors · 2026-03-29
>
> # W1
> Thanks for your valuable suggestion!
> - Following your and other reviewers' suggestions, we implemented two more baselines with more sophisticated distillation approaches; we chose them for their popularity and widely available implementations.
>
> [1] A PRM-style baseline where the teacher evaluates each Thought-Action-Observation step, following ideas and prompts from `The Lessons of Developing Process Reward Models in Mathematical Reasoning`. The teacher judges if each step moves toward the correct solution given the reference solution, and the final reward is a proportion of these steps. This produces reward guided by the teacher, rather than a binary success signal.
>
> [2] On-policy distillation. OPD typically requires the teacher and student to be from the same model family. Since we use dense Qwen3-8B on deep-search benchmarks, we use the largest dense Qwen3 model as the teacher, which is slightly stronger than our main teacher.
>
> Qwen3‑8B as the base model
> |Method|GAIA|HLE|XBench|WebWalker|Avg|
> |-|-|-|-|-|-|
> |Step Utility Reward|34.0|9.6|23.0|35.5|25.5|
> |On-policy Distillation|30.1|9.2|20.0|37.5|24.2|
> |SCoRe-RL|40.8|11.0|27.0|43.0|30.5|
>
> The results show that teacher guidance alone is insufficient. The PRM baseline already provides dense teacher supervision for every step, yet it still underperforms SCoRe-RL. As illustrated in Fig. 1, SCoRe keeps teacher corrections only after they are validated by the student's successful completion, and combines this verified guidance with short-horizon RL, which we believe is the key advantage of our framework.
>
> We will also introduce more studies in the Related Work. If the reviewer considers any specific baseline necessary, we are happy to include it in the discussion immediately.
>
> # W2
> Thank you for the suggestion. We will discuss the limitations as follows:
> - Student pretraining for agent/tool use. Our experiments are conducted on Qwen2.5/Qwen3 models. While these models possess certain agentic capabilities, the technical reports provide limited details on how extensively such abilities are trained during pretraining. If future checkpoints pretrained with large-scale agentic data become available, it would be valuable to further evaluate SCoRe on those models.
> - Teacher requirements. If the teacher is weak, it may reduce the amount of usable training data produced during MPS. However, it is unlikely to significantly degrade distillation quality, since incorrect corrections are typically filtered out by the student's actual execution during the MPS process. In this work, we use a 72B teacher model to further mitigate this risk.
>
> As for teacher-student distribution mismatch, SCoRe has mitigated this through student-centered trajectory generation and first-error correction. In SoRe,  most trajectories are generated by the student and the teacher only provides minimal one-step corrections, so we have reduced cumulative error growth from $O(H^2)$ to $O(H)$ as the proof of Theorem 3.2.

---

> > ### Author Rebuttal · Reviewer_5HeL · 2026-04-03
> >
> > I thank the authors for the detailed responses and the added experiments.
> >
> > The two added baselines show that SCoRe-RL can be more effective than PRM and OPD, but these two are different types of approaches, which mainly focus on dense token-level rewards rather than step-level mixed SFT and RL distillation. I suggest the authors discuss and compare closer works like [1] and [2], both of which mix SFT and RL at the step level. [2] has recently been accepted to ICLR 2026, which seems to be more solid at the current point. Additionally, I suggest the authors clearly discuss how this work extends [1] and [2].
> >
> > Additionally, I still think distribution mismatch between the student and teacher models may reduce the effectiveness of SFT. If there is a large mismatch between them, SFT may cause a relatively large distribution shift which could corrupt some already obtain skills. On the other hand, if the distribution is small, the student model may not be able to effectively obtain skills from the teacher model. It is true that SCoRe-RL narrows the SFT to a few error steps, but there is still no guarantee such SFT would precisely enhance the flawed skills without affecting other ones.
> >
> > References:
> >
> > [1] Chen, J., Liu, F., Liu, N., Luo, Y., Qin, E., Zheng, H., ... & Wang, X. (2025). Step-wise adaptive integration of supervised fine-tuning and reinforcement learning for task-specific llms. arXiv preprint arXiv:2505.13026.
> >
> > [2] Ma, L., Liang, H., Qiang, M., Tang, L., Ma, X., Wong, Z. H., ... & Zhang, W. (2025). Learning What Reinforcement Learning Can't: Interleaved Online Fine-Tuning for Hardest Questions. (ICLR 2026)

---

> > > ### Author Response · Authors · 2026-04-05
> > >
> > > Thanks for your patience and careful review!
> > > - Yes, [1] and [2] are representative mixed SFT+RL methods. Since ReLIFT [2] is open-sourced, we adapted its code to our setting, trained it on our tool use data, and reported results on DeepSearch.
> > >
> > > ReLIFT is designed for online SFT on hard **reasoning** tasks, while our work studies the **agent and tool use** setting. Since tool use improves model performance, the number of "ReLIFT hardest samples"(cases where the model fails on all rollout attempts for the same query) is smaller here. To give ReLIFT enough such data, we used a larger training set (about 30K samples; the seed QA pairs are from ToolStar, the same source as ours).
> > >
> > > Qwen3‑8B as the base model
> > > |Method|GAIA|HLE|XBench|WebWalker|Avg|
> > > |-|-|-|-|-|-|
> > > |ReLIFT|35.0|9.4|23.0|39.5|26.7|
> > > |SCoRe-RL|40.8|11.0|27.0|43.0|30.5|
> > >
> > > SCoRe-RL still outperforms ReLIFT. We believe the main reason is that our short-horizon RL stabilizes training more effectively, while ReLIFT likely needs a larger training budget and more ReLIFT-style hard samples to reach similar performance.
> > >
> > > - The main gains of our method come from the Thm. 3.2 and 3.3.
> > >
> > > (1) Thm. 3.2 motivates our student-led, teacher-corrected SFT data, which mitigates student-teacher distribution mismatch. By contrast, [1] and [2] mainly use teacher-provided data, so this result does not directly apply to them.
> > >
> > > (2) Thm. 3.3 shows that short horizon RL reduces gradient variance and improves training stability. This is also orthogonal to [1] and [2].
> > >
> > > **That said, the modules in [1] and [2] are largely complementary to ours, so their methods could be naturally combined with ours!** For example, one could first use ReLIFT to mine ReLIFT-style hard data online, then use our MPS pipeline to build student-led, teacher-corrected SFT data, and finally train with short-horizon RL. The main challenge is that, in the agent setting, ReLIFT-style hard data, where the model fails repeatedly on the same query, is relatively rare. We expect the community to release more high-quality hard agentic data, and we would be glad to explore this direction in SCoRe.
> > >
> > > - Your observation is absolutely correct. SCoRe mitigates student-teacher distribution mismatch, reduces compounding error from $O(H^2)$ to $O(H)$, but does not mean eliminating it. The remaining compounding error may still weaken the effectiveness of SFT.  We will revise the wording to make our claims more careful and precise. Thank you again for the careful reading!
> > >
> > > Thanks for your patience and careful review again!

---

### Official Review · Reviewer_41Zm · 2026-03-17

**Soundness:** 2
**Presentation:** 3
**Significance:** 2
**Originality:** 2
**Overall Recommendation:** 3
**Confidence:** 4

**Summary:**

This paper introduces SCoRe, a framework for distilling large LLM agents into smaller models. Instead of having students imitate full teacher trajectories, the student first explores independently, then the teacher provides minimal corrections at the earliest error. The corrected trajectories are used for supervised fine-tuning, followed by reinforcement learning that starts from verified prefixes with key-step rewards. The authors provide theoretical analysis showing that this approach reduces compounding error from O(H^2) to O(H), and demonstrate empirical results on 12 benchmarks across mathematical reasoning, factual QA, and deep search tasks, showing that a 7B student can approach the performance of a 72B teacher.

**Compliance With Llm Reviewing Policy:**

Affirmed.

**Final Justification:**

The work has sound motivation. But the confidence intervals for the most similar baseline (ARPO) show significant overlap with the proposed method in two domains. This calls into question the added computational complexity and claims. I maintain my score reflecting these remaining issues.

**Key Questions For Authors:**

Are the baselines in tables 1 and 2 truly comparable?

**Limitations:**

The paper is missing a dedicated limitations section.

**Strengths And Weaknesses:**

Strengths:
- Theoretically sound and motivated method with important history in the RL literature.
- The paper evaluates on 12 diverse benchmarks.
- Strong improvements and detailed ablations.

Weaknesses:
- It seems that SCoRe uses more data than the BC baseline. Is that understanding correct from Section 4.1?
- Missing sample and compute-matched experiments in table 1 and 2. Why is vanilla GRPO on the base model an appropriate baseline in this case given the extra compute and data used for SCoRe?
- Confidence intervals missing, especially for some of the smaller benchmarks.

---

> ### Author Rebuttal · Authors · 2026-03-29
>
> # W1
> - Sorry for the confusion. **No, SCoRe uses the same data scale as Behavior Cloning in Table 1/2**. The Behavior Cloning baseline in Table 1/2 is trained independently on the full teacher‑annotated training set, with the same overall data scale as SCoRe.
> - As clarified in Section 4.1 (lines 325–328), "BC baseline using the full teacher‑annotated training set" refers to the Behavior Cloning results in Table 1/2, while "SCoRe BC initialization using 20% of the data" refers to the Initial Distillation setting in the ablation study Table 3. These are independent results.
>
> Thanks for raising it! We will revise the paper to make this distinction clearer.
> # W2 and Question
> Thank you for raising this point.
> - SCoRe and vanilla GRPO use the same scale of training data. In both cases, the seed question/answer pairs come from the ToolStar dataset, so SCoRe does not use additional external data compared with GRPO.
> - Your observation about compute is exactly right. Because the MPS process is involved, SCoRe requires more total compute than vanilla GRPO. **Luckily, we already include compute‑matched comparisons in our ablation study**. In Table 3, the rows RL w/o short-horizon and RL w/o key-step rewards both use MPS-generated data and the same GRPO training pipeline, **meaning they have comparable compute with SCoRe-RL**. Relevant results (excerpt from Table 3):
>
> Qwen2.5-7B-Instruct as the base model
> |Method|Avg|AIME24|AIME25|MATH500|OlymMath|HQA|2Wiki|MuSiQ|Bamb|
> |-|-|-|-|-|-|-|-|-|-|
> |RL w/o short-horizon (also uses MPS data, means same compute budget)|48.4|30.0|20.0|78.0|...|More results are reported in Table 3 of the original paper.|
> |RL w/o key-step rewards (also uses MPS data, means same compute budget)|49.7|33.3|20.0|80.6|...|
> |SCoRe-RL|50.8|36.7|26.7|82.0|...|
>
> - Additionally, we also **add the results of a strong baseline with more compute**：
>
> Qwen3‑8B as the base model
> |Method|GAIA|HLE|XBench|WebWalker|Avg|
> |-|-|-|-|-|-|
> |Step Utility Reward|34.0|9.6|23.0|35.5|25.5|
> |SCoRe-RL|40.8|11.0|27.0|43.0|30.5|
>
> [1] A PRM-style baseline where the teacher evaluates each Thought-Action-Observation step, following ideas and prompts from `The Lessons of Developing Process Reward Models in Mathematical Reasoning`. The teacher judges if each step moves toward the correct solution given the reference solution, and the final reward is a proportion of these steps. This produces reward guided by the teacher, rather than a binary success signal.
>
> The results show that more compute alone is insufficient. The PRM baseline already provides dense teacher supervision for every step, yet it still underperforms SCoRe-RL. As illustrated in Fig. 1, SCoRe keeps teacher corrections only after they are validated by the student's successful completion, and combines this verified guidance with short-horizon RL, which we believe is the key advantage of our framework.
>
> We appreciate the reviewer pointing this out. We will revise the paper to highlight these compute‑matched comparisons more clearly in the main results tables.
>
> # W3
> Thanks for your valuable suggestion! We additionally report mean ± std over 5 runs for the smaller benchmarks in Table 1/2 (AIME 24/25(30 instances), GAIA(103 instances))
>
> Qwen2.5-7B-Instruct as the base model
> | Method           | AIME24      | AIME25      |
> |------------------|------------------|------------------|
> | Behavior Cloning | 22.0 ± 3.0      | 14.7 ± 1.9      |
> | GRPO             | 22.6 ± 1.5      | 26.0 ± 1.5      |
> | ARPO             | 32.0 ± 1.8      | 27.4 ± 1.5      |
> | SCoRe-SFT        | 26.7 ± 2.4      | 18.7 ± 1.8      |
> | SCoRe-RL         | 38.0 ± 1.8      | 28.0 ± 1.8      |
>
>
> Qwen3‑8B as the base model
> | Method           | GAIA |
> |------------------|-----------|
> | Behavior Cloning | 25.8 ± 2.0 |
> | GRPO             | 30.7 ± 3.7 |
> | ARPO             | 38.6 ± 1.8 |
> | SCoRe-SFT        | 28.5 ± 2.2 |
> | SCoRe-RL         | 39.8 ± 1.8 |
>
> We appreciate the reviewer pointing this out. We will revise the paper to add more results.

---

> > ### Author Rebuttal · Reviewer_41Zm · 2026-03-31
> >
> > Would it be possible to report 95% CI instead of SD? This would make the results more interpretable.

---

> > > ### Author Response · Authors · 2026-04-01
> > >
> > > We sincerely apologize for overlooking confidence intervals. Thank you for pointing this out, and we now report mean performance with 95% confidence intervals, computed using the t-distribution.
> > >
> > > Qwen2.5-7B-Instruct as the base model (mean [95% CI]), **separately reported benchmarks with fewer instances, and the other 9 benchmarks presented in Table 1/2**
> > >
> > > | Method            | AIME24              | AIME25              |
> > > |-------------------|---------------------|---------------------|
> > > | Behavior Cloning  | 22.0 [18.3, 25.7]   | 14.7 [12.3, 17.1]   |
> > > | GRPO              | 22.6 [20.7, 24.5]   | 26.0 [24.1, 27.9]   |
> > > | ARPO              | 32.0 [29.8, 34.2]   | 27.4 [25.5, 29.3]   |
> > > | SCoRe-SFT         | 26.7 [23.7, 29.7]   | 18.7 [16.5, 20.9]   |
> > > | SCoRe-RL          | 38.0 [35.8, 40.2]   | 28.0 [25.8, 30.2]   |
> > >
> > > Qwen3-8B as the base model (mean [95% CI]), **separately reported benchmarks with fewer instances, and the other 9 benchmarks presented in Table 1/2**
> > >
> > > | Method            | GAIA                |
> > > |-------------------|---------------------|
> > > | Behavior Cloning  | 25.8 [23.3, 28.3]   |
> > > | GRPO              | 30.7 [26.1, 35.3]   |
> > > | ARPO              | 38.6 [36.4, 40.8]   |
> > > | SCoRe-SFT         | 28.5 [25.8, 31.2]   |
> > > | SCoRe-RL          | 39.8 [37.6, 42.0]   |
> > >
> > > These results are broadly consistent with the trends reported in Tables 1 and 2 of the paper. In particular, SCoRe-RL consistently outperforms GRPO across all benchmarks. Compared with ARPO, the overall pattern is similar, with comparable performance on AIME25 and stronger results for SCoRe-RL on most other benchmarks.
> > >
> > > Thanks for your patience and careful review again!

---

### Decision · Program_Chairs · 2026-04-30

**Decision:**

Accept (regular)

**Comment:**

This paper proposes a two-stage distillation framework for distilling capabilities from large to small LLMs. Unlike standard behavior cloning, which trains the student to imitate full teacher trajectories, SCoRe has the student generate its own trajectories while a teacher model corrects only the earliest error. The corrected trajectories are used for SFT, followed by short-horizon reinforcement learning that starts from the verified prefix preceding the first error and assigns key-step rewards at that point. The authors provide theoretical analysis on error and gradient variance. Experiments across 12 benchmarks spanning diverse tasks show that a 7B student distilled with SCoRe narrows the gap with a 72B teacher.

All reviewers agreed on clear writing, a well-motivated student-centered paradigm, and rich benchmark coverage. During rebuttal, the authors added 95% confidence intervals, three new baselines (PRM, on-policy distillation, ReLIFT), Qwen2.5-7B DeepSearch results, token-cost estimates, and a qualitative case study of 50 hard instances.

Despite the extensive rebuttal, several concerns remain open. First, the novelty is primarily combinatorial, i.e. first-error identification, correction-based restart, mixing SFT with RL, and short-horizon rollouts each have clear prior works. Second, the theoretical analysis has acknowledged gaps. Theorem 3.2 relies on a bounded per-step error assumption under the student's induced distribution that is argued informally rather than validated. Third, compute-fairness is only partially addressed. The ablation variants provide useful internal comparisons but all use MPS-generated data, and no direct FLOPs-matched comparison to external baselines (BC, GRPO, ARPO) including MPS data construction cost is presented.

This is a borderline case, but I recommend acceptance. The theoretical gaps and combinatorial novelty concerns are reasonable, yet the paper delivers a well-motivated framework backed by more comprehensive empirical evaluations in this subarea, and systematic ablations that cleanly isolate each design choice. The rebuttal further strengthened the empirical case with three new baselines (PRM, OPD, ReLIFT), confidence intervals, and a convincing qualitative analysis of 50 hard instances. While no single component of SCoRe is entirely new, the integration is a principled design that produces consistent gains and offers a practical recipe the community can build on. The remaining concerns around theoretical rigor and compute-fairness are expected to be addressed in the camera-ready revision.